# Learning Space-Time Continuous Neural PDEs from Partially Observed States

**Valerii Iakovlev**   **Markus Heinonen**   **Harri Lähdesmäki**
Department of Computer Science, Aalto University, Finland
{valerii.iakovlev, markus.o.heinonen, harri.lahdesmaki}@aalto.fi

## Abstract

We introduce a novel grid-independent model for learning partial differential equations (PDEs) from noisy and partial observations on irregular spatiotemporal grids. We propose a space-time continuous latent neural PDE model with an efficient probabilistic framework and a novel encoder design for improved data efficiency and grid independence. The latent state dynamics are governed by a PDE model that combines the collocation method and the method of lines. We employ amortized variational inference for approximate posterior estimation and utilize a multiple shooting technique for enhanced training speed and stability. Our model demonstrates state-of-the-art performance on complex synthetic and real-world datasets, overcoming limitations of previous approaches and effectively handling partially-observed data. The proposed model outperforms recent methods, showing its potential to advance data-driven PDE modeling and enabling robust, grid-independent modeling of complex partially-observed dynamic processes.

## 1 Introduction

Modeling spatiotemporal processes allows to understand and predict the behavior of complex systems that evolve over time and space (Cressie and Wikle, 2011). Partial differential equations (PDEs) are a popular tool for this task as they have a solid mathematical foundation (Evans, 2010) and can describe the dynamics of a wide range of physical, biological, and social phenomena (Murray, 2002; Hirsch, 2007). However, deriving PDEs can be challenging, especially when the system's underlying mechanisms are complex and not well understood. Data-driven methods can bypass these challenges (Brunton and Kutz, 2019). By learning the underlying system dynamics directly from data, we can develop accurate PDE models that capture the essential features of the system. This approach has changed our ability to model complex systems and make predictions about their behavior in a data-driven manner.

While current data-driven PDE models have been successful at modeling complex spatiotemporal phenomena, they often operate under various simplifying assumptions such as regularity of the spatial or temporal grids (Long et al., 2018; Kochkov et al., 2021; Pfaff et al., 2021; Li et al., 2021; Han et al., 2022; Poli et al., 2022), discreteness in space or time (Seo et al., 2020; Pfaff et al., 2021; Lienen and Günnemann, 2022; Brandstetter et al., 2022), and availability of complete and noiseless observations (Long et al., 2018; Pfaff et al., 2021; Wu et al., 2022). Such assumptions become increasingly limiting in more realistic scenarios with scarce data and irregularly spaced, noisy and partial observations.

We address the limitations of existing methods and propose a space-time continuous and grid-independent model that can learn PDE dynamics from noisy and partial observations made on irregular spatiotemporal grids. Our main contributions include:

---

Source code and datasets can be found in our [github repository](#).

37th Conference on Neural Information Processing Systems (NeurIPS 2023).

- Development of an efficient generative modeling framework for learning latent neural PDE models from noisy and partially-observed data;
- Novel PDE model that merges two PDE solution techniques – the collocation method and the method of lines – to achieve space-time continuity, grid-independence, and data efficiency;
- Novel encoder design that operates on local spatiotemporal neighborhoods for improved data-efficiency and grid-independence.

Our model demonstrates state-of-the-art performance on complex synthetic and real-world datasets, opening up the possibility for accurate and efficient modeling of complex dynamic processes and promoting further advancements in data-driven PDE modeling.

## 2 Problem Setup

In this work we are concerned with modeling of spatiotemporal processes. For brevity, we present our method for a single observed trajectory, but extension to multiple trajectories is straightforward. We observe a spatiotemporal dynamical system evolving over time on a spatial domain $\Omega$. The observations are made at $M$ arbitrary consecutive time points $t_{1:M} := (t_1, \ldots, t_M)$ and $N$ arbitrary observation locations $\boldsymbol{x}_{1:N} := (\boldsymbol{x}_1, \ldots, \boldsymbol{x}_N)$, where $\boldsymbol{x}_i \in \Omega$. This generates a sequence of observations $\boldsymbol{u}_{1:M} := (\boldsymbol{u}_1, \ldots, \boldsymbol{u}_M)$, where $\boldsymbol{u}_i \in \mathbb{R}^{N \times D}$ contains $D$-dimensional observations at the $N$ observation locations. We define $\boldsymbol{u}_i^j$ as the observation at time $t_i$ and location $\boldsymbol{x}_j$. The number of time points and observation locations may vary between different observed trajectories.

We assume the data is generated by a dynamical system with a latent state $\boldsymbol{z}(t, \boldsymbol{x}) \in \mathbb{R}^d$, where $t$ is time and $\boldsymbol{x} \in \Omega$ is spatial location. The latent state is governed by an unknown PDE and is mapped to the observed state $\boldsymbol{u}(t, \boldsymbol{x}) \in \mathbb{R}^D$ by an unknown observation function $g$ and likelihood model $p$:

$$\frac{\partial \boldsymbol{z}(t, x)}{\partial t} = F(\boldsymbol{z}(t, \boldsymbol{x}), \partial_{\boldsymbol{x}} \boldsymbol{z}(t, \boldsymbol{x}), \partial_{\boldsymbol{x}}^2 \boldsymbol{z}(t, \boldsymbol{x}), \ldots), \tag{1}$$

$$\boldsymbol{u}(t, \boldsymbol{x}) \sim p(g(\boldsymbol{z}(t, \boldsymbol{x}))), \tag{2}$$

where $\partial_{\boldsymbol{x}}^\bullet \boldsymbol{z}(t, \boldsymbol{x})$ denotes partial derivatives wrt $\boldsymbol{x}$.

In this work we make two assumptions that are highly relevant in real-world scenarios. First, we assume partial observations, that is, the observed state $\boldsymbol{u}(t, \boldsymbol{x})$ does not contain all information about the latent state $\boldsymbol{z}(t, \boldsymbol{x})$ (e.g., $\boldsymbol{z}(t, \boldsymbol{x})$ contains pressure and velocity, but $\boldsymbol{u}(t, \boldsymbol{x})$ contains information only about the pressure). Second, we assume out-of-distribution time points and observation locations, that is, their number, positions, and density can change arbitrarily at test time.

## 3 Model

Here we describe the model components (Sec. 3.1) which are then used to construct the generative model (Sec. 3.2).

### 3.1 Model components

Our model consists of four parts: space-time continuous latent state $\boldsymbol{z}(t, \boldsymbol{x})$ and observed state $\boldsymbol{u}(t, \boldsymbol{x})$, a dynamics function $F_{\theta_{\text{dyn}}}$ governing the temporal evolution of the latent state, and an observation function $g_{\theta_{\text{dec}}}$ mapping the latent state to the observed state (see Figure 1). Next, we describe these components in detail.

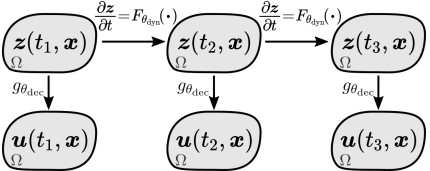

Figure 1: Model sketch. Initial latent state $\boldsymbol{z}(t_1, \boldsymbol{x})$ is evolved via $F_{\theta_{\text{dyn}}}$ to the following latent states which are then mapped to the observed states by $g_{\theta_{\text{dec}}}$.

**Latent state.** To define a space-time continuous latent state $\boldsymbol{z}(t, \boldsymbol{x}) \in \mathbb{R}^d$, we introduce $\boldsymbol{z}(t) := (\boldsymbol{z}^1(t), \ldots, \boldsymbol{z}^N(t)) \in \mathbb{R}^{N \times d}$, where each $\boldsymbol{z}^i(t) \in \mathbb{R}^d$ corresponds to the observation location $\boldsymbol{x}_i$. Then, we define the latent state $\boldsymbol{z}(t, \boldsymbol{x})$ as a spatial interpolant of $\boldsymbol{z}(t)$:

$$\boldsymbol{z}(t, \boldsymbol{x}) := \text{Interpolate}(\boldsymbol{z}(t))(\boldsymbol{x}), \tag{3}$$

where $\text{Interpolate}(\cdot)$ maps $\boldsymbol{z}(t)$ to an interpolant which can be evaluated at any spatial location $\boldsymbol{x} \in \Omega$ (see Figure 2). We do not rely on a particular interpolation method, but in this work we use linear interpolation as it shows good performance and facilitates efficient implementation.

**Latent state dynamics.** Given a space-time continuous latent state, one can naturally define its dynamics in terms of a PDE:

$$\frac{\partial z(t, x)}{\partial t} = F_{\theta_{\text{dyn}}}(z(t, x), \partial_x z(t, x), \partial_x^2 z(t, x), \ldots), \qquad (4)$$

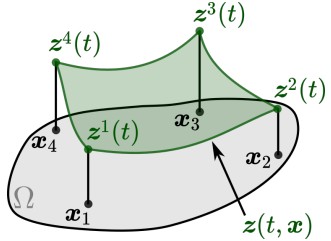

Figure 2: Latent state $z(t, x)$ defined as an interpolant of $z(t) := (z^1(t), ..., z^4(t))$.

where $F_{\theta_{\text{dyn}}}$ is a dynamics function with parameters $\theta_{\text{dyn}}$. This is a viable approach known as the collocation method (Kansa, 1990; Cheng, 2009), but it has several limitations. It requires us to decide which partial derivatives to include in the dynamics function, and also requires an interpolant which has all the selected partial derivatives (e.g., linear interpolant has only first order derivatives). To avoid these limitations, we combine the collocation method with another PDE solution technique known as the method of lines (Schiesser, 1991; Hamdi et al., 2007), which approximates spatial derivatives $\partial_x^\bullet z(t, x)$ using only evaluations of $z(t, x)$, and then let the dynamics function approximate all required derivatives in a data-driven manner. To do that, we define the spatial neighborhood of $x$ as $\mathcal{N}_S(x)$, which is a set containing $x$ and its spatial neighbors, and also define $z(t, \mathcal{N}_S(x))$, which is a set of evaluations of the interpolant $z(t, x)$ at points in $\mathcal{N}_S(x)$:

$$\mathcal{N}_S(x) := \{x' \in \Omega : x' = x \text{ or } x' \text{ is a spatial neighbor of } x\}, \qquad (5)$$
$$z(t, \mathcal{N}_S(x)) := \{z(t, x') : x' \in \mathcal{N}_S(x)\}, \qquad (6)$$

and assume that this information is sufficient to approximate all required spatial derivatives at $x$. This is a reasonable assumption since, e.g., finite differences can approximate derivatives using only function values and locations of the evaluation points. Hence, we define the dynamics of $z(t, x)$ as

$$\frac{\partial z(t, x)}{\partial t} = F_{\theta_{\text{dyn}}}(\mathcal{N}_S(x), z(t, \mathcal{N}_S(x))), \qquad (7)$$

which is defined only in terms of the values of the latent state, but not its spatial derivatives.

One way to define the spatial neighbors for $x$ is in terms of the observation locations $x_{1:N}$ (e.g., use the nearest ones) as was done, for example, in (Long et al., 2018; Pfaff et al., 2021; Lienen and Günnemann, 2022). Instead, we utilize continuity of the latent state $z(t, x)$, and define the spatial neighbors in a grid-independent manner as a fixed number of points arranged in a predefined patter around $x$ (see Figure 3). This allows to fix the shape and size of the spatial neighborhoods in advance, making them independent of the observation locations. In this work we use the spatial neighborhood consisting of two concentric circles of radius $r$ and $r/2$, each circle contains 8 evaluation points as in Figure 3. In Appendix D we compare neighborhoods of various shapes and sizes.

Figure 3: Example of $\mathcal{N}_S(x_i)$. Instead of using the observation locations (dots) to define spatial neighbors, we use spatial locations arranged in a fixed predefined pattern (crosses).

Equation 7 allows to simulate the temporal evolution of $z(t, x)$ at any spatial location. However, since $z(t, x)$ is defined only in terms of a spatial interpolant of $z(t)$ (see Eq. 3), with $z^i(t) = z(t, x_i)$, it is sufficient to simulate the latent state dynamics only at the observation locations $x_{1:N}$. Hence, we can completely characterize the latent state dynamics in terms of a system of $N$ ODEs:

$$\frac{dz(t)}{dt} := \begin{pmatrix} \frac{dz^1(t)}{dt} \\ \vdots \\ \frac{dz^N(t)}{dt} \end{pmatrix} = \begin{pmatrix} \frac{\partial z(t, x_1)}{\partial t} \\ \vdots \\ \frac{\partial z(t, x_N)}{\partial t} \end{pmatrix} = \begin{pmatrix} F_{\theta_{\text{dyn}}}(\mathcal{N}_S(x_1), z(t, \mathcal{N}_S(x_1))) \\ \vdots \\ F_{\theta_{\text{dyn}}}(\mathcal{N}_S(x_N), z(t, \mathcal{N}_S(x_N))) \end{pmatrix}. \qquad (8)$$

For convenience, we define $z(t; t_1, z_1, \theta_{\text{dyn}}) := \text{ODESolve}(t; t_1, z_1, \theta_{\text{dyn}})$ as the solution of the ODE system in Equation 8 at time $t$ with initial state $z(t_1) = z_1$ and parameters $\theta_{\text{dyn}}$. We also define $z(t, x; t_1, z_1, \theta_{\text{dyn}})$ as the spatial interpolant of $z(t; t_1, z_1, \theta_{\text{dyn}})$ as in Equation 3. We solve the ODEs using off the shelf differentiable ODE solvers from torchdiffeq package (Chen, 2018). Note that we solve for the state $z(t)$ only at the observation locations $x_{1:N}$, so to get the neighborhood values $z(t, \mathcal{N}_S(x_i))$ we perform interpolation at every step of the ODE solver.

**Observation function.** We define the mapping from the latent space to the observation space as a parametric function $g_{\theta_{dec}}$ with parameters $\theta_{dec}$:

$$\boldsymbol{u}(t, \boldsymbol{x}) \sim \mathcal{N}(g_{\theta_{dec}}(\boldsymbol{z}(t, \boldsymbol{x})), \sigma_u^2 I_D), \tag{9}$$

where $\mathcal{N}$ is the Gaussian distribution, $\sigma_u^2$ is noise variance, and $I_D$ is $D$-by-$D$ identity matrix.

## 3.2 Generative model

Training models of dynamic systems is often challenging due to long training times and training instabilities (Ribeiro et al., 2020; Metz et al., 2021). To alleviate these problems, various heuristics have been proposed, such as progressive lengthening and splitting of the training trajectories (Yildiz et al., 2019; Um et al., 2020). We use multiple shooting (Bock and Plitt, 1984; Voss et al., 2004), a simple and efficient technique which has demonstrated its effectiveness in ODE learning applications (Jordana et al., 2021; Hegde et al., 2022). We extent the multiple shooting framework for latent ODE models presented in (Iakovlev et al., 2023) to our PDE modeling setup by introducing spatial dimensions in the latent state and designing an encoder adapted specifically to the PDE setting (Section 4.2).

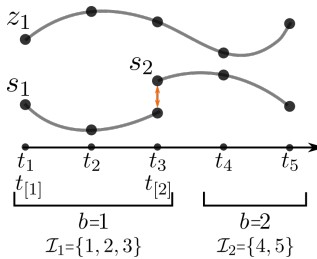

Figure 4: Multiple shooting splits a trajectory with one initial state (top) into two subtrajectories with two initial states (bottom) and tries to minimize the gap between subtrajectories (orange arrow).

Multiple shooting splits a single trajectory $\{\boldsymbol{z}(t_i)\}_{i=1,...,M}$ with one initial state $\boldsymbol{z}_1$ into $B$ consecutive non-overlapping subtrajectories $\{\boldsymbol{z}(t_i)\}_{i \in \mathcal{I}_b}$, $b = 1, \ldots, B$ with $B$ initial states $\boldsymbol{s}_{1:B} := (\boldsymbol{s}_1, \ldots, \boldsymbol{s}_B)$ while imposing a continuity penalty between the subtrajectories (see Figure 4). The index set $\mathcal{I}_b$ contains time point indices for the $b$'th sub-trajectory. We also denote the temporal position of $\boldsymbol{s}_b$ as $t_{[b]}$ and place $\boldsymbol{s}_b$ at the first time point preceding the $b$'th sub-trajectory (except $\boldsymbol{s}_1$ which is placed at $t_1$). Note that the shooting states $\boldsymbol{s}_b$ have the same dimension as the original latent state $\boldsymbol{z}(t)$ i.e., $\boldsymbol{s}_b \in \mathbb{R}^{N \times d}$. Multiple shooting allows to parallelize the simulation over the sub-trajectories and shortens the simulation intervals thus improving the training speed and stability. In Appendix D we demonstrate the effect of multiple shooting on the model training and prediction accuracy.

We begin by defining the prior over the unknown model parameters and initial states:

$$p(\boldsymbol{s}_{1:B}, \theta_{dyn}, \theta_{dec}) = p(\boldsymbol{s}_{1:B}|\theta_{dyn})p(\theta_{dyn})p(\theta_{dec}), \tag{10}$$

where $p(\theta_{dyn})$ and $p(\theta_{dec})$ are zero-mean diagonal Gaussians, and the continuity inducing prior $p(\boldsymbol{s}_{1:B}|\theta_{dyn})$ is defined as in (Iakovlev et al., 2023)

$$p(\boldsymbol{s}_{1:B}|\theta_{dyn}) = p(\boldsymbol{s}_1) \prod_{b=2}^{B} p(\boldsymbol{s}_b|\boldsymbol{s}_{b-1}, \theta_{dyn}). \tag{11}$$

Intuitively, the continuity prior $p(\boldsymbol{s}_b|\boldsymbol{s}_{b-1}, \theta_{dyn})$ takes the initial latent state $\boldsymbol{s}_{b-1}$, simulates it forward from time $t_{[b-1]}$ to $t_{[b]}$ to get $\boldsymbol{\mu}_{[b]} = \text{ODESolve}(t_{[b]}; t_{[b-1]}, \boldsymbol{s}_{b-1}, \theta_{dyn})$, and then forces $\boldsymbol{\mu}_{[b]}$ to approximately match the initial state $\boldsymbol{s}_b$ of the next sub-trajectory, thus promoting continuity of the full trajectory. We assume the continuity inducing prior factorizes across the grid points, i.e.,

$$p(\boldsymbol{s}_{1:B}|\theta_{dyn}) = \left[ \prod_{j=1}^{N} p(\boldsymbol{s}_1^j) \right] \left[ \prod_{b=2}^{B} \prod_{j=1}^{N} p(\boldsymbol{s}_b^j|\boldsymbol{s}_{b-1}, \theta_{dyn}) \right], \tag{12}$$

$$= \left[ \prod_{j=1}^{N} p(\boldsymbol{s}_1^j) \right] \left[ \prod_{b=2}^{B} \prod_{j=1}^{N} \mathcal{N}\left( \boldsymbol{s}_b^j | \boldsymbol{z}(t_{[b]}, \boldsymbol{x}_j; t_{[b-1]}, \boldsymbol{s}_{b-1}, \theta_{dyn}), \sigma_c^2 I_d \right) \right], \tag{13}$$

where $p(\boldsymbol{s}_1^j)$ is a diagonal Gaussian, and parameter $\sigma_c^2$ controls the strength of the prior. Note that the term $\boldsymbol{z}(t_{[b]}, \boldsymbol{x}_j; t_{[b-1]}, \boldsymbol{s}_{b-1}, \theta_{dyn})$ in Equation 13 equals the ODE forward solution $\text{ODESolve}(t_{[b]}; t_{[b-1]}, \boldsymbol{s}_{b-1}, \theta_{dyn})$ at grid location $\boldsymbol{x}_j$.

Finally, we define our generative in terms of the following sampling procedure:

$$\theta_{\text{dyn}}, \theta_{\text{dec}}, \boldsymbol{s}_{1:B} \sim p(\theta_{\text{dyn}})p(\theta_{\text{dec}})p(\boldsymbol{s}_{1:B}|\theta_{\text{dyn}}), \tag{14}$$

$$\boldsymbol{z}(t_i) = \boldsymbol{z}(t_i; t_{[b]}, \boldsymbol{s}_b, \theta_{\text{dyn}}), \qquad\qquad b \in \{1, ..., B\},\ i \in \mathcal{I}_b, \tag{15}$$

$$\boldsymbol{u}_i^j \sim p(\boldsymbol{u}_i^j|g_{\theta_{\text{dec}}}(\boldsymbol{z}(t_i, \boldsymbol{x}_j)), \qquad\qquad i = 1, \dots, M,\ j = 1, \dots, N, \tag{16}$$

with the following joint distribution (see Appendix A for details about the model specification.):

$$p(\boldsymbol{u}_{1:M}, \boldsymbol{s}_{1:B}, \theta_{\text{dyn}}, \theta_{\text{dec}}) = \prod_{b=1}^{B} \prod_{i \in \mathcal{I}_b} \prod_{j=1}^{N} \left[ p(\boldsymbol{u}_i^j|\boldsymbol{s}_b, \theta_{\text{dyn}}, \theta_{\text{dec}}) \right] p(\boldsymbol{s}_{1:B}|\theta_{\text{dyn}})p(\theta_{\text{dyn}})p(\theta_{\text{dec}}). \tag{17}$$

# 4 Parameter Inference, Encoder, and Forecasting

## 4.1 Amortized variational inference

We approximate the true posterior over the model parameters and initial states $p(\boldsymbol{s}_{1:B}, \theta_{\text{dyn}}, \theta_{\text{dec}}|\boldsymbol{u}_{1:M})$ using variational inference (Blei et al., 2017) with the following approximate posterior:

$$q(\theta_{\text{dyn}}, \theta_{\text{dec}}, \boldsymbol{s}_{1:B}) = q(\theta_{\text{dyn}})q(\theta_{\text{dec}})q(\boldsymbol{s}_{1:B}) = q_{\boldsymbol{\psi}_{\text{dyn}}}(\theta_{\text{dyn}})q_{\boldsymbol{\psi}_{\text{dec}}}(\theta_{\text{dec}}) \prod_{b=1}^{B} \prod_{j=1}^{N} q_{\boldsymbol{\psi}_b^j}(\boldsymbol{s}_b^j), \tag{18}$$

where $q_{\boldsymbol{\psi}_{\text{dyn}}}$, $q_{\boldsymbol{\psi}_{\text{dec}}}$ and $q_{\boldsymbol{\psi}_b^j}$ are diagonal Gaussians, and $\boldsymbol{\psi}_{\text{dyn}}$, $\boldsymbol{\psi}_{\text{dec}}$ and $\boldsymbol{\psi}_b^j$ are variational parameters. To avoid direct optimization over the local variational parameters $\boldsymbol{\psi}_b^j$, we use amortized variational inference (Kingma and Welling, 2013) and train an encoder $h_{\theta_{\text{enc}}}$ with parameters $\theta_{\text{enc}}$ which maps observations $\boldsymbol{u}_{1:M}$ to $\boldsymbol{\psi}_b^j$ (see Section 4.2). For brevity, we sometimes omit the dependence of approximate posteriors on variational parameters and simply write e.g., $q(\boldsymbol{s}_b^j)$.

In variational inference the best approximation of the posterior is obtained by minimizing the Kullback-Leibler divergence: $\text{KL}\big[q(\theta_{\text{dyn}}, \theta_{\text{dec}}, \boldsymbol{s}_{1:B})\|p(\theta_{\text{dyn}}, \theta_{\text{dec}}, \boldsymbol{s}_{1:B}|\boldsymbol{u}_{1:N})\big]$, which is equivalent to maximizing the evidence lower bound (ELBO), defined for our model as:

$$\mathcal{L} = \underbrace{\sum_{b=1}^{B} \sum_{i \in \mathcal{I}_b} \sum_{j=1}^{N} \mathbb{E}_{q(\boldsymbol{s}_b, \theta_{\text{dyn}}, \theta_{\text{dec}})} \left[ \log p(\boldsymbol{u}_i^j|\boldsymbol{s}_b, \theta_{\text{dyn}}, \theta_{\text{dec}}) \right]}_{(i)\ \textit{observation model}} - \underbrace{\sum_{j=1}^{N} \text{KL}\big[q(\boldsymbol{s}_1^j)\|p(\boldsymbol{s}_1^j)\big]}_{(ii)\ \textit{initial state prior}}$$

$$- \underbrace{\sum_{b=2}^{B} \sum_{j=1}^{N} \mathbb{E}_{q(\theta_{\text{dyn}}, \boldsymbol{s}_{b-1})} \left[ \text{KL}\big[q(\boldsymbol{s}_b^j)\|p(\boldsymbol{s}_b^j|\boldsymbol{s}_{b-1}, \theta_{\text{dyn}})\big] \right]}_{(iii)\ \textit{continuity prior}} - \underbrace{\text{KL}\big[q(\theta_{\text{dyn}})\|p(\theta_{\text{dyn}})\big]}_{(iv)\ \textit{dynamics prior}} - \underbrace{\text{KL}\big[q(\theta_{\text{dec}})\|p(\theta_{\text{dec}})\big]}_{(v)\ \textit{decoder prior}}.$$

The terms $(ii)$, $(iv)$, and $(v)$ are computed analytically, while terms $(i)$ and $(iii)$ are approximated using Monte Carlo integration for expectations, and numerical ODE solvers for initial value problems. See Appendix A and B approximate posterior details and derivation and computation of the ELBO.

## 4.2 Encoder

Here we describe our encoder which maps observations $\boldsymbol{u}_{1:M}$ to local variational parameters $\boldsymbol{\psi}_b^j$ required to sample the initial latent state of the sub-trajectory $b$ at time point $t_{[b]}$ and observation location $\boldsymbol{x}_j$. Similarly to our model, the encoder should be data-efficient and grid-independent.

Similarly to our model (Section 3.1), we enable grid-independence by making the encoder operate on spatial interpolants of the observations $\boldsymbol{u}_{1:M}$ (even if they are noisy):

$$\boldsymbol{u}_i(\boldsymbol{x}) := \text{Interpolate}(\boldsymbol{u}_i)(\boldsymbol{x}), \qquad i = 1, \dots, M, \tag{19}$$

where spatial interpolation is done separately for each time point $i$. We then use the interpolants $\boldsymbol{u}_i(\boldsymbol{x})$ to define the spatial neighborhoods $\mathcal{N}_{\text{S}}(\boldsymbol{x})$ in a grid-independent manner.

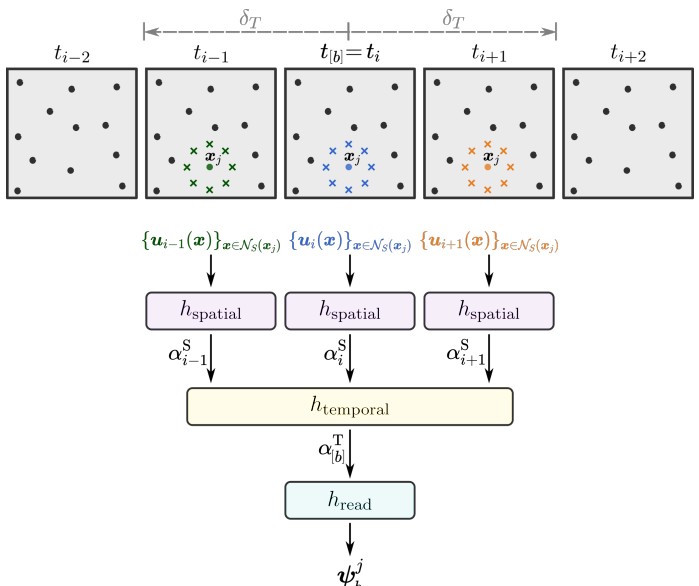

Figure 5: Spatiotemporal neighborhood of a multiple shooting time point $t_{[b]} = t_i$ and location $\boldsymbol{x}_j$, $\boldsymbol{u}[t_{[b]}, \boldsymbol{x}_j]$ (denoted by green, blue and orange crosses and the dots inside), is mapped to the variational parameters $\boldsymbol{\psi}_b^j$ via the encoder.

To improve data-efficiency, we assume $\boldsymbol{\psi}_b^j$ does not depend on the whole observed sequence $\boldsymbol{u}_{1:M}$, but only on some local information in a spatiotemporal neighborhood of $t_{[b]}$ and $\boldsymbol{x}_j$. We define the temporal neighborhood of $t_{[b]}$ as

$$\mathcal{N}_{\mathrm{T}}(t_{[b]}) \coloneqq \{k : |t_k - t_{[b]}| \leq \delta_T, \ k = 1, \dots, M\}, \tag{20}$$

where $\delta_T$ is a hyperparameter controlling the neighborhood size, and then define the spatiotemporal neighborhood of $t_{[b]}$ and $\boldsymbol{x}_j$ as

$$\boldsymbol{u}[t_{[b]}, \boldsymbol{x}_j] \coloneqq \{\boldsymbol{u}_k(\boldsymbol{x}) : k \in \mathcal{N}_{\mathrm{T}}(t_{[b]}), \boldsymbol{x} \in \mathcal{N}_{\mathrm{S}}(\boldsymbol{x}_j)\}. \tag{21}$$

Our encoder operates on such spatiotemporal neighborhoods $\boldsymbol{u}[t_{[b]}, \boldsymbol{x}_j]$ and works in three steps (see Figure 5). First, for each time index $k \in \mathcal{N}_{\mathrm{T}}(t_{[b]})$ it aggregates the spatial information $\{\boldsymbol{u}_k(\boldsymbol{x})\}_{\boldsymbol{x} \in \mathcal{N}(\boldsymbol{x}_j)}$ into a vector $\alpha_k^{\mathrm{S}}$. Then, it aggregates the spatial representations $\alpha_k^{\mathrm{S}}$ across time into another vector $\alpha_{[b]}^{\mathrm{T}}$ which is finally mapped to the variational parameters $\boldsymbol{\psi}_b^j$ as follows:

$$\boldsymbol{\psi}_b^j = h_{\theta_{\mathrm{enc}}}(\boldsymbol{u}[t_{[b]}, \boldsymbol{x}_j]) = h_{\mathrm{read}}(h_{\mathrm{temporal}}(h_{\mathrm{spatial}}(\boldsymbol{u}[t_{[b]}, \boldsymbol{x}_j]))). \tag{22}$$

**Spatial aggregation.** Since the spatial neighborhoods are fixed and remain identical for all spatial locations (see Figure 5), we implement the spatial aggregation function $h_{\mathrm{spatial}}$ as an MLP which takes elements of the set $\{\boldsymbol{u}_k(\boldsymbol{x})\}_{\boldsymbol{x} \in \mathcal{N}_{\mathrm{S}}(\boldsymbol{x}_j)}$ stacked in a fixed order as the input.

**Temporal aggregation.** We implement $h_{\mathrm{temporal}}$ as a stack of transformer layers (Vaswani et al., 2017) which allows it to operate on input sets of arbitrary size. We use time-aware attention and continuous relative positional encodings (Iakovlev et al., 2023) which were shown to be effective on data from dynamical systems observed at irregular time intervals. Each transformer layer takes a layer-specific input set $\{\xi_k^{\mathrm{in}}\}_{k \in \mathcal{N}_{\mathrm{T}}(t_{[b]})}$, where $\xi_k^{\mathrm{in}}$ is located at $t_k$, and maps it to an output set $\{\xi_k^{\mathrm{out}}\}_{k \in \mathcal{N}_{\mathrm{T}}(t_{[b]})}$, where each $\xi_k^{\mathrm{out}}$ is computed using only the input elements within distance $\delta_T$ from $t_k$, thus promoting temporal locality. Furthermore, instead of using absolute positional encodings the model assumes the behavior of the system does not depend on time and uses relative temporal distances to inject positional information. The first layer takes $\{\alpha_k^{\mathrm{S}}\}_{k \in \mathcal{N}_{\mathrm{T}}(t_{[b]})}$ as the input, while the last layer returns a single element at time point $t_{[b]}$, which represents the temporal aggregation $\alpha_{[b]}^{\mathrm{T}}$.

**Variational parameter readout.** Since $\alpha_i^{\mathrm{T}}$ is a fixed-length vector, we implement $h_{\mathrm{read}}$ as an MLP.

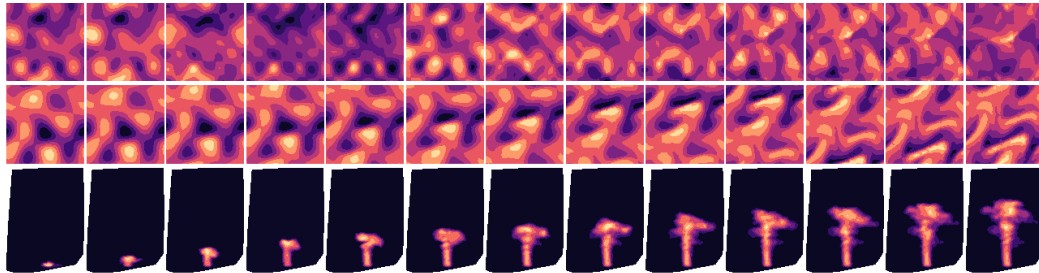

Figure 6: **Top:** SHALLOW WATER dataset contains observations of the wave height in a pool of water. **Middle:** NAVIER-STOKES dataset contains observations of the concentration of a species transported in a fluid via buoyancy and velocity field. **Bottom:** SCALAR FLOW dataset contains observations of smoke plumes raising in warm air.

### 4.3 Forecasting

Given initial observations $\tilde{\boldsymbol{u}}_{1:m}$ at time points $t_{1:m}$, we predict the future observation $\tilde{\boldsymbol{u}}_n$ at a time point $t_n > t_m$ as the expected value of the approximate posterior predictive distribution:

$$p(\tilde{\boldsymbol{u}}_n|\tilde{\boldsymbol{u}}_{1:m}, \boldsymbol{u}_{1:M}) \approx \int p(\tilde{\boldsymbol{u}}_n|\tilde{\boldsymbol{s}}_m, \theta_{\text{dyn}}, \theta_{\text{dec}})q(\tilde{\boldsymbol{s}}_m)q(\theta_{\text{dyn}})q(\theta_{\text{dec}})d\tilde{\boldsymbol{s}}_m d\theta_{\text{dyn}}d\theta_{\text{dec}}. \quad (23)$$

The expected value is estimated via Monte Carlo integration (see Appendix C.4 for details).

## 5   Experiments

We use three challenging datasets: SHALLOW WATER, NAVIER-STOKES, and SCALAR FLOW which contain observations of spatiotemporal system at $N \approx 1100$ grid points evolving over time (see Figure 6). The first two datasets are synthetic and generated using numeric PDE solvers (we use scikit-fdiff (Cellier, 2019) for SHALLOW WATER, and PhiFlow (Holl et al., 2020) for NAVIER-STOKES), while the third dataset contains real-world observations (camera images) of smoke plumes raising in warm air (Eckert et al., 2019). In all cases the observations are made at irregular spatiotemporal grids and contain only partial information about the true system state. In particular, for SHALLOW WATER we observe only the wave height, for NAVIER-STOKES we observe only the concentration of the species, and for SCALAR FLOW only pixel densities are known. All datasets contain 60/20/20 training/validation/testing trajectories. See Appendix C for details.

We train our model for 20k iterations with constant learning rate of 3e-4 and linear warmup. The latent spatiotemporal dynamics are simulated using differentiable ODE solvers from the torchdiffeq package (Chen, 2018) (we use dopri5 with rtol=1e-3, atol=1e-4, no adjoint). Training is done on a single NVIDIA Tesla V100 GPU, with a single run taking 3-4 hours. We use the mean absolute error (MAE) on the test set as the performance measure. Error bars are standard errors over 4 random seeds. For forecasting we use the expected value of the posterior predictive distribution. See Appendix C for all details about the training, validation, and testing setup.

**Latent state dimension.**   Here we show the advantage of using latent-space models on partially observed data. We change the latent state dimension $d$ from 1 to 5 and measure the test MAE. Note that for $d = 1$ we effectively have a data-space model which models the observations without trying to reconstruct the missing states. Figure 7 shows that in all cases there is improvement in performance as the latent dimension grows. For SHALLOW WATER and NAVIER-STOKES the true latent dimension is 3. Since SCALAR FLOW is a real-world process, there is no true latent dimension. As a benchmark, we provide the performance of our model trained on fully-observed versions of the synthetic datasets (we use the same architecture and hyperparameters, but fix $d$ to 3). Figure 7 also shows examples of model predictions (at the final time point) for different values of $d$. We see a huge difference between $d = 1$ and $d = 3, 5$. Note how apparently small difference in MAE at $d = 1$ and $d = 5$ for SCALAR FLOW corresponds to a dramatic improvement in the prediction quality.

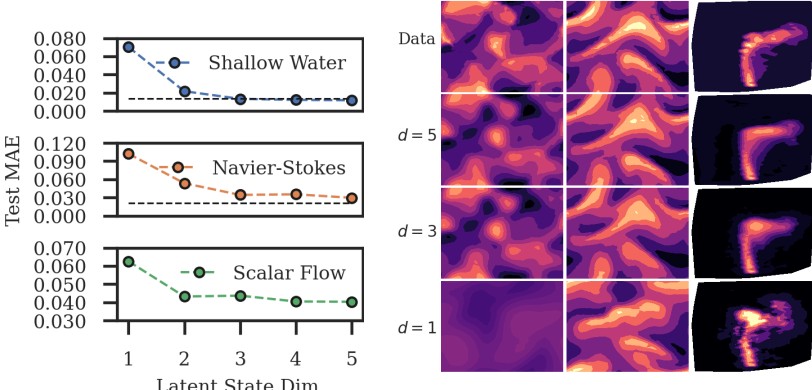

Figure 7: **Left:** Test MAE vs latent state dimension $d$. Black lines are test MAE on fully-observed versions of the datasets ($\pm$ standard error). **Right:** Model predictions for different $d$.

**Grid independence.** In this experiment we demonstrate the grid-independence property of our model by training it on grids with $\approx 1100$ observation locations, and then testing on a coarser, original, and finer grids. We evaluate the effect of using different interpolation methods by repeating the experiment with linear, k-nearest neighbors, and inverse distance weighting (IDW) interpolants. For SHALLOW WATER and NAVIER-STOKES the coarser/finer grids contain 290/4200 nodes, while for SCALAR FLOW we have 560/6420 nodes, respectively. Table 1 shows the model's performance for different spatial grids and interpolation methods. We see that all interpolation methods perform rather similarly on the original grid, but linear interpolation and IDW tend to perform better on finer/coarser grids than k-NN. Performance drop on coarse grids is expected since we get less accurate information about the system's initial state and simulate the dynamics on coarse grids. Figure 8 also shows examples of model predictions (at the final time point) for different grid sizes and linear interpolant.

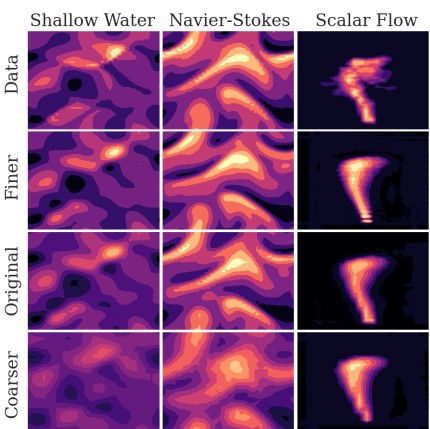

Figure 8: Predictions on spatial grids of different density (linear interpolant, test data).

**Comparison to other models.** We test our model against recent spatiotemporal models from the literature: Finite Element Networks (FEN) (Lienen and Günnemann, 2022), Neural Stochastic PDEs (NSPDE) (Salvi et al., 2021), MAgNet (Boussif et al., 2022), and DINo (Yin et al., 2023). We also use Neural ODEs (NODE) (Chen et al., 2018) as the baseline. We use the official implementation for all models and tune their hyperparameters for the best performance (see App.

Table 1: Test MAE for different interpolation methods.

| Dataset | Grid | k-NN | Linear | IDW |
|---|---|---|---|---|
| Shallow Water | Coarser | $0.046 \pm 0.002$ | $0.034 \pm 0.001$ | $0.038 \pm 0.002$ |
| | Original | $0.017 \pm 0.002$ | $0.016 \pm 0.002$ | $0.017 \pm 0.003$ |
| | Finer | $0.031 \pm 0.003$ | $0.017 \pm 0.003$ | $0.030 \pm 0.002$ |
| Navier Stokes | Coarser | $0.087 \pm 0.006$ | $0.069 \pm 0.009$ | $0.066 \pm 0.006$ |
| | Original | $0.048 \pm 0.009$ | $0.041 \pm 0.003$ | $0.045 \pm 0.010$ |
| | Finer | $0.054 \pm 0.009$ | $0.044 \pm 0.004$ | $0.049 \pm 0.002$ |
| Scalar Flow | Coarser | $0.041 \pm 0.021$ | $0.032 \pm 0.009$ | $0.035 \pm 0.012$ |
| | Original | $0.019 \pm 0.001$ | $0.018 \pm 0.000$ | $0.018 \pm 0.001$ |
| | Finer | $0.040 \pm 0.016$ | $0.026 \pm 0.006$ | $0.028 \pm 0.007$ |

C for details). For SHALLOW WATER and NAVIER-STOKES we use the first 5 time points to infer the latent state and then predict the next 20 time points, while for SCALAR FLOW we use the first 10 points for inference and predict the next 10 points. For synthetic data, we consider two settings: one where the data is fully observed (i.e., the complete state is recorded) – a setting for which most models are designed – and one where the data is partially observed (i.e., only part of the full state

is given, as discussed at the beginning of this section). The results are shown in Table 2. We see that some of the baseline models achieve reasonably good results on the fully-observed datasets, but they fail on partially-observed data, while our model maintains strong performance in all cases. Apart from the fully observed SHALLOW WATER dataset where FEN performs slightly better, our method outperforms other methods on all other datasets by a clear margin. See Appendix C for hyperparameter details. In Appendix E we demonstrate our model's capability to learn dynamics from noisy data. In Appendix F we show model predictions on different datasets.

Table 2: Test MAE for different models.

| Model | Shallow Water (Full) | Shallow Water (Partial) | Navier Stokes (Full) | Navier Stokes (Partial) | Scalar Flow |
|---|---|---|---|---|---|
| NODE | $0.036 \pm 0.000$ | $0.084 \pm 0.001$ | $0.053 \pm 0.001$ | $0.109 \pm 0.001$ | $0.056 \pm 0.001$ |
| FEN | $\mathbf{0.011 \pm 0.002}$ | $0.064 \pm 0.005$ | $0.031 \pm 0.001$ | $0.108 \pm 0.002$ | $0.062 \pm 0.005$ |
| SNPDE | $0.019 \pm 0.002$ | $0.033 \pm 0.001$ | $0.042 \pm 0.004$ | $0.075 \pm 0.002$ | $0.059 \pm 0.002$ |
| DINo | $0.027 \pm 0.001$ | $0.063 \pm 0.003$ | $0.047 \pm 0.001$ | $0.113 \pm 0.002$ | $0.059 \pm 0.001$ |
| MAgNet | NA | $0.061 \pm 0.001$ | NA | $0.103 \pm 0.003$ | $0.056 \pm 0.003$ |
| Ours | $0.014 \pm 0.002$ | $\mathbf{0.016 \pm 0.002}$ | $\mathbf{0.024 \pm 0.003}$ | $\mathbf{0.041 \pm 0.003}$ | $\mathbf{0.042 \pm 0.001}$ |

## 6 Related Work

Closest to our work is Ayed et al. (2022), where they considered the problem of learning PDEs from partial observations and proposed a discrete and grid-dependent model that is restricted to regular spatiotemporal grids. Another related work is that of Nguyen et al. (2020), where they proposed a variational inference framework for learning ODEs from noisy and partially-observed data. However, they consider only low-dimensional ODEs and are restricted to regular grids.

Other works considered learning the latent space PDE dynamics using the "encode-process-decode" approach. Pfaff et al. (2021) use GNN-based encoder and dynamics function and map the observations to the same spatial grid in the latent space and learn the latent space dynamics. Sanchez et al. (2020) use a similar approach but with CNNs and map the observations to a coarser latent grid and learn the coarse-scale dynamics. Wu et al. (2022) use CNNs to map observations to a low-dimensional latent vector and learn the latent dynamics. However, all these approaches are grid-dependent, limited to regular spatial/temporal grids, and require fully-observed data.

Interpolation has been used in numerous studies for various applications. Works such as (Alet et al., 2019; Jiang et al., 2020; Han et al., 2022) use interpolation to map latent states on coarse grids to observations on finer grids. Hua et al. (2022) used interpolation as a post-processing step to obtain continuous predictions, while Boussif et al. (2022) used it to recover observations at missing nodes.

Another approach for achieving grid-independence was presented in neural operators (Li et al., 2021; Lu et al., 2021), which learn a mapping between infinite-dimensional function spaces and represent the mapping in a grid-independent manner.

## 7 Conclusion

We proposed a novel space-time continuous, grid-independent model for learning PDE dynamics from noisy and partial observations on irregular spatiotemporal grids. Our contributions include an efficient generative modeling framework, a novel latent PDE model merging collocation and method of lines, and a data-efficient, grid-independent encoder design. The model demonstrates state-of-the-art performance on complex datasets, highlighting its potential for advancing data-driven PDE modeling and enabling accurate predictions of spatiotemporal phenomena in diverse fields. However, our model and encoder operate on every spatial and temporal location which might not be the most efficient approach and hinders scaling to extremely large grids, hence research into more efficient latent state extraction and dynamics modeling methods is needed.

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

# A  Appendix A

## A.1  Model specification.

Here we provide all details about our model specification. The joint distribution for our model is

$$p(\boldsymbol{u}_{1:M}, \boldsymbol{s}_{1:B}, \theta_{\text{dyn}}, \theta_{\text{dec}}) = p(\boldsymbol{u}_{1:N}|\boldsymbol{s}_{1:B}, \theta_{\text{dyn}}, \theta_{\text{dec}})p(\boldsymbol{s}_{1:B}|\theta_{\text{dyn}})p(\theta_{\text{dyn}})p(\theta_{\text{dec}}). \tag{24}$$

Next, we specify each component in detail.

**Parameter priors.**   The parameter priors are isotropic zero-mean multivariate normal distributions:

$$p(\theta_{\text{dyn}}) = \mathcal{N}(\theta_{\text{dyn}}|\mathbf{0}, I), \tag{25}$$
$$p(\theta_{\text{dec}}) = \mathcal{N}(\theta_{\text{dec}}|\mathbf{0}, I), \tag{26}$$

where $\mathcal{N}$ is the normal distribution, $\mathbf{0}$ is a zero vector, and $I$ is the identity matrix, both have an appropriate dimensionality dependent on the number of encoder and dynamics parameters.

**Continuity prior.**   We define the continuity prior as

$$p(\boldsymbol{s}_{1:B}|\theta_{\text{dyn}}) = p(\boldsymbol{s}_1)\prod_{b=2}^{B} p(\boldsymbol{s}_b|\boldsymbol{s}_{b-1}, \theta_{\text{dyn}}), \tag{27}$$

$$= \left[\prod_{j=1}^{N} p(\boldsymbol{s}_1^j)\right]\left[\prod_{b=2}^{B}\prod_{j=1}^{N} p(\boldsymbol{s}_b^j|\boldsymbol{s}_{b-1}, \theta_{\text{dyn}})\right], \tag{28}$$

$$= \left[\prod_{j=1}^{N} \mathcal{N}(\boldsymbol{s}_1^j|\mathbf{0}, I)\right]\left[\prod_{b=2}^{B}\prod_{j=1}^{N} \mathcal{N}\left(\boldsymbol{s}_b^j|\boldsymbol{z}(t_{[b]}, \boldsymbol{x}_j; t_{[b-1]}, \boldsymbol{s}_{b-1}, \theta_{\text{dyn}}), \sigma_c^2 I\right)\cdot\right], \tag{29}$$

where $\mathcal{N}$ is the normal distribution, $\mathbf{0} \in \mathbb{R}^d$ is a zero vector, $I \in \mathbb{R}^{d\times d}$ is the identity matrix, and $\sigma_c \in \mathbb{R}$ is the parameter controlling the strength of the prior. Smaller values of $\sigma_c$ tend to produce smaller gaps between the sub-trajectories.

**Observation model**

$$p(\boldsymbol{u}_{1:N}|\boldsymbol{s}_{1:B}, \theta_{\text{dyn}}, \theta_{\text{dec}}) = \prod_{b=1}^{B}\prod_{i\in\mathcal{I}_b}\prod_{j=1}^{N} p(\boldsymbol{u}_i^j|\boldsymbol{s}_b, \theta_{\text{dyn}}, \theta_{\text{dec}}) \tag{30}$$

$$= \prod_{b=1}^{B}\prod_{i\in\mathcal{I}_b}\prod_{j=1}^{N} p(\boldsymbol{u}_i^j|g_{\theta_{\text{dec}}}(\boldsymbol{z}(t_i, \boldsymbol{x}_j; t_{[b]}, \boldsymbol{s}_b, \theta_{\text{dyn}}))) \tag{31}$$

$$= \prod_{b=1}^{B}\prod_{i\in\mathcal{I}_b}\prod_{j=1}^{N} \mathcal{N}(\boldsymbol{u}_i^j|g_{\theta_{\text{dec}}}(\boldsymbol{z}(t_i, \boldsymbol{x}_j; t_{[b]}, \boldsymbol{s}_b, \theta_{\text{dyn}})), \sigma_u^2 I), \tag{32}$$

where $\mathcal{N}$ is the normal distribution, $\sigma_u^2$ is the observation noise variance, and $I \in \mathbb{R}^{D\times D}$ is the identity matrix. Note again that $\boldsymbol{z}(t_i, \boldsymbol{x}_j; t_{[b]}, \boldsymbol{s}_b, \theta_{\text{dyn}})$ above equals the ODE forward solution ODESolve$(t_i; t_{[b]}, \boldsymbol{s}_b, \theta_{\text{dyn}})$ at grid location $\boldsymbol{x}_j$.

## A.2  Approximate posterior specification.

Here we provide all details about the approximate posterior. We define the approximate posterior as

$$q(\theta_{\text{dyn}}, \theta_{\text{dec}}, \boldsymbol{s}_{1:B}) = q(\theta_{\text{dyn}})q(\theta_{\text{dec}})q(\boldsymbol{s}_{1:B}) = q_{\psi_{\text{dyn}}}(\theta_{\text{dyn}})q_{\psi_{\text{dec}}}(\theta_{\text{dec}})\prod_{b=1}^{B}\prod_{j=1}^{N} q_{\psi_b^j}(\boldsymbol{s}_b^j). \tag{33}$$

Next, we specify each component in detail.

**Dynamics parameters posterior.** We define $q_{\psi_{\text{dyn}}}(\theta_{\text{dyn}})$ as

$$q_{\psi_{\text{dyn}}}(\theta_{\text{dyn}}) = \mathcal{N}(\theta_{\text{dyn}}|\gamma_{\text{dyn}}, \text{diag}(\tau_{\text{dyn}}^2)), \tag{34}$$

where $\gamma_{\text{dyn}}$ and $\tau_{\text{dyn}}^2$ are vectors with an appropriate dimension (dependent on the number of dynamics parameters), and $\text{diag}(\tau_{\text{dyn}}^2)$ is a matrix with $\tau_{\text{dyn}}^2$ on the diagonal. We define the vector of variational parameters as $\psi_{\text{dyn}} = (\gamma_{\text{dyn}}, \tau_{\text{dyn}}^2)$. We optimize directly over $\psi_{\text{dyn}}$ and initialize $\gamma_{\text{dyn}}$ using Xavier (Glorot and Bengio, 2010) initialization, while $\tau_{\text{dyn}}$ is initialized with each element equal to $9 \cdot 10^{-4}$.

**Decoder parameters posterior.** We define $q_{\psi_{\text{dec}}}(\theta_{\text{dec}})$ as

$$q_{\psi_{\text{dec}}}(\theta_{\text{dec}}) = \mathcal{N}(\theta_{\text{dec}}|\gamma_{\text{dec}}, \text{diag}(\tau_{\text{dec}}^2)), \tag{35}$$

where $\gamma_{\text{dec}}$ and $\tau_{\text{dec}}^2$ are vectors with an appropriate dimension (dependent on the number of decoder parameters), and $\text{diag}(\tau_{\text{dec}}^2)$ is a matrix with $\tau_{\text{dec}}^2$ on the diagonal. We define the vector of variational parameters as $\psi_{\text{dec}} = (\gamma_{\text{dec}}, \tau_{\text{dec}}^2)$. We optimize directly over $\psi_{\text{dec}}$ and initialize $\gamma_{\text{dec}}$ using Xavier (Glorot and Bengio, 2010) initialization, while $\tau_{\text{dec}}$ is initialized with each element equal to $9 \cdot 10^{-4}$.

**Shooting variables posterior.** We define $q_{\psi_b^j}(s_b^j)$ as

$$q_{\psi_b^j}(s_b^j) = \mathcal{N}(s_b^j|\gamma_b^j, \text{diag}([\tau_b^j]^2))), \tag{36}$$

where the vectors $\gamma_b^j, \tau_b^j \in \mathbb{R}^d$ are returned by the encoder $h_{\theta_{\text{enc}}}$, and $\text{diag}([\tau_b^j]^2)$ is a matrix with $[\tau_b^j]^2$ on the diagonal. We define the vector of variational parameters as $\psi_b^j = (\gamma_b^j, [\tau_b^j])$. Because the variational inference for the shooting variables is amortized, our model is trained w.r.t. the parameters of the encoder network, $\theta_{\text{enc}}$.

# B Appendix B

### B.1 Derivation of ELBO.

For our model and the choice of the approximate posterior the ELBO can be written as

$$\mathcal{L} = \int q(\theta_{\text{dyn}}, \theta_{\text{dec}}, s_{1:B}) \ln \frac{p(u_{1:M}, s_{1:B}, \theta_{\text{dyn}}, \theta_{\text{dec}})}{q(\theta_{\text{dyn}}, \theta_{\text{dec}}, s_{1:B})} d\theta_{\text{dyn}}d\theta_{\text{dec}}ds_{1:B} \tag{37}$$

$$= \int q(\theta_{\text{dyn}}, \theta_{\text{dec}}, s_{1:B}) \ln \frac{p(u_{1:M}|s_{1:B}, \theta_{\text{dyn}}, \theta_{\text{dec}})p(s_{1:B}|\theta_{\text{dyn}})p(\theta_{\text{dyn}})p(\theta_{\text{dec}})}{q(s_{1:B})q(\theta_{\text{dyn}})q(\theta_{\text{dec}})} d\theta_{\text{dyn}}d\theta_{\text{dec}}ds_{1:B} \tag{38}$$

$$= \int q(\theta_{\text{dyn}}, \theta_{\text{dec}}, s_{1:B}) \ln p(u_{1:M}|s_{1:B}, \theta_{\text{dyn}}, \theta_{\text{dec}})d\theta_{\text{dyn}}d\theta_{\text{dec}}ds_{1:B} \tag{39}$$

$$- \int q(\theta_{\text{dyn}}, \theta_{\text{dec}}, s_{1:B}) \ln \frac{q(s_{1:B})}{p(s_{1:B}|\theta_{\text{dyn}})}d\theta_{\text{dyn}}d\theta_{\text{dec}}ds_{1:B} \tag{40}$$

$$- \int q(\theta_{\text{dyn}}, \theta_{\text{dec}}, s_{1:B}) \ln \frac{q(\theta_{\text{dyn}})}{p(\theta_{\text{dyn}})}d\theta_{\text{dyn}}d\theta_{\text{dec}}ds_{1:B} \tag{41}$$

$$- \int q(\theta_{\text{dec}}, \theta_{\text{dec}}, s_{1:B}) \ln \frac{q(\theta_{\text{dec}})}{p(\theta_{\text{dec}})}d\theta_{\text{dyn}}d\theta_{\text{dec}}ds_{1:B} \tag{42}$$

$$= \mathcal{L}_1 - \mathcal{L}_2 - \mathcal{L}_3 - \mathcal{L}_4. \tag{43}$$

Next, we will look at each term $\mathcal{L}_i$ separately.

$$\mathcal{L}_1 = \int q(\theta_{\text{dyn}}, \theta_{\text{dec}}, \boldsymbol{s}_{1:B}) \ln p(\boldsymbol{u}_{1:M} | \boldsymbol{s}_{1:B}, \theta_{\text{dyn}}, \theta_{\text{dec}}) d\theta_{\text{dyn}} d\theta_{\text{dec}} d\boldsymbol{s}_{1:B} \tag{44}$$

$$= \int q(\theta_{\text{dyn}}, \theta_{\text{dec}}, \boldsymbol{s}_{1:B}) \ln \left[ \prod_{b=1}^{B} \prod_{i \in \mathcal{I}_b} \prod_{j=1}^{N} p(\boldsymbol{u}_i^j | \boldsymbol{s}_b, \theta_{\text{dyn}}, \theta_{\text{dec}}) \right] d\theta_{\text{dyn}} d\theta_{\text{dec}} d\boldsymbol{s}_{1:B} \tag{45}$$

$$= \sum_{b=1}^{B} \sum_{i \in \mathcal{I}_b} \sum_{j=1}^{N} \int q(\theta_{\text{dyn}}, \theta_{\text{dec}}, \boldsymbol{s}_{1:B}) \ln \left[ p(\boldsymbol{u}_i^j | \boldsymbol{s}_b, \theta_{\text{dyn}}, \theta_{\text{dec}}) \right] d\theta_{\text{dyn}} d\theta_{\text{dec}} d\boldsymbol{s}_{1:B} \tag{46}$$

$$= \sum_{b=1}^{B} \sum_{i \in \mathcal{I}_b} \sum_{j=1}^{N} \int q(\theta_{\text{dyn}}, \theta_{\text{dec}}, \boldsymbol{s}_b) \ln \left[ p(\boldsymbol{u}_i^j | \boldsymbol{s}_b, \theta_{\text{dyn}}, \theta_{\text{dec}}) \right] d\theta_{\text{dyn}} d\theta_{\text{dec}} d\boldsymbol{s}_b \tag{47}$$

$$= \sum_{b=1}^{B} \sum_{i \in \mathcal{I}_b} \sum_{j=1}^{N} \mathbb{E}_{q(\theta_{\text{dyn}}, \theta_{\text{dec}}, \boldsymbol{s}_b)} \ln \left[ p(\boldsymbol{u}_i^j | \boldsymbol{s}_b, \theta_{\text{dyn}}, \theta_{\text{dec}}) \right]. \tag{48}$$

$$\mathcal{L}_2 = \int q(\theta_{\text{dyn}}, \theta_{\text{dec}}, \boldsymbol{s}_{1:B}) \ln \frac{q(\boldsymbol{s}_{1:B})}{p(\boldsymbol{s}_{1:B} | \theta_{\text{dyn}})} d\theta_{\text{dyn}} d\theta_{\text{dec}} d\boldsymbol{s}_{1:B} \tag{49}$$

$$= \int q(\theta_{\text{dyn}}, \theta_{\text{dec}}, \boldsymbol{s}_{1:B}) \ln \left[ \frac{q(\boldsymbol{s}_1)}{p(\boldsymbol{s}_1)} \prod_{b=2}^{B} \frac{q(\boldsymbol{s}_b)}{p(\boldsymbol{s}_b | \boldsymbol{s}_{b-1}, \theta_{\text{dyn}})} \right] d\theta_{\text{dyn}} d\theta_{\text{dec}} d\boldsymbol{s}_{1:B} \tag{50}$$

$$= \int q(\theta_{\text{dyn}}, \theta_{\text{dec}}, \boldsymbol{s}_{1:B}) \ln \left[ \prod_{j=1}^{N} \frac{q(\boldsymbol{s}_1^j)}{p(\boldsymbol{s}_1^j)} \right] d\theta_{\text{dyn}} d\theta_{\text{dec}} d\boldsymbol{s}_{1:B} \tag{51}$$

$$+ \int q(\theta_{\text{dyn}}, \theta_{\text{dec}}, \boldsymbol{s}_{1:B}) \ln \left[ \prod_{b=2}^{B} \prod_{j=1}^{N} \frac{q(\boldsymbol{s}_b^j)}{p(\boldsymbol{s}_b^j | \boldsymbol{s}_{b-1}, \theta_{\text{dyn}})} \right] d\theta_{\text{dyn}} d\theta_{\text{dec}} d\boldsymbol{s}_{1:B} \tag{52}$$

$$= \sum_{j=1}^{N} \int q(\theta_{\text{dyn}}, \theta_{\text{dec}}, \boldsymbol{s}_{1:B}) \ln \left[ \frac{q(\boldsymbol{s}_1^j)}{p(\boldsymbol{s}_1^j)} \right] d\theta_{\text{dyn}} d\theta_{\text{dec}} d\boldsymbol{s}_{1:B} \tag{53}$$

$$+ \sum_{b=2}^{B} \int q(\theta_{\text{dyn}}, \theta_{\text{dec}}, \boldsymbol{s}_{1:B}) \sum_{j=1}^{N} \ln \left[ \frac{q(\boldsymbol{s}_b^j)}{p(\boldsymbol{s}_b^j | \boldsymbol{s}_{b-1}, \theta_{\text{dyn}})} \right] d\theta_{\text{dyn}} d\theta_{\text{dec}} d\boldsymbol{s}_{1:B} \tag{54}$$

$$= \sum_{j=1}^{N} \int q(\boldsymbol{s}_1^j) \ln \left[ \frac{q(\boldsymbol{s}_1^j)}{p(\boldsymbol{s}_1^j)} \right] d\boldsymbol{s}_1^j \tag{55}$$

$$+ \sum_{b=2}^{B} \int q(\theta_{\text{dyn}}, \boldsymbol{s}_{b-1}, \boldsymbol{s}_b) \sum_{j=1}^{N} \ln \left[ \frac{q(\boldsymbol{s}_b^j)}{p(\boldsymbol{s}_b^j | \boldsymbol{s}_{b-1}, \theta_{\text{dyn}})} \right] d\theta_{\text{dyn}} d\boldsymbol{s}_{b-1} d\boldsymbol{s}_b \tag{56}$$

$$= \sum_{j=1}^{N} \int q(\boldsymbol{s}_1^j) \ln \left[ \frac{q(\boldsymbol{s}_1^j)}{p(\boldsymbol{s}_1^j)} \right] d\boldsymbol{s}_1^j \tag{57}$$

$$+ \sum_{b=2}^{B} \int q(\theta_{\text{dyn}}, \boldsymbol{s}_{b-1}) \sum_{j=1}^{N} \left[ \int q(\boldsymbol{s}_b^j) \ln \frac{q(\boldsymbol{s}_b^j)}{p(\boldsymbol{s}_b^j | \boldsymbol{s}_{b-1}, \theta_{\text{dyn}})} d\boldsymbol{s}_b^j \right] d\theta_{\text{dyn}} d\boldsymbol{s}_{b-1} \tag{58}$$

$$= \sum_{j=1}^{N} \text{KL} \left( q(\boldsymbol{s}_1^j) \| p(\boldsymbol{s}_1^j) \right) + \sum_{b=2}^{B} \mathbb{E}_{q(\theta_{\text{dyn}}, \boldsymbol{s}_{b-1})} \left[ \sum_{j=1}^{N} \text{KL} \left( q(\boldsymbol{s}_b^j) \| p(\boldsymbol{s}_b^j | \boldsymbol{s}_{b-1}, \theta_{\text{dyn}}) \right) \right], \tag{59}$$

where KL is Kullback–Leibler (KL) divergence. Both of the KL divergences above have a closed form but the expectation w.r.t. $q(\theta_{\text{dyn}}, \boldsymbol{s}_{b-1})$ does not.

$$\mathcal{L}_3 = \text{KL}(q(\theta_{\text{dyn}}) \| p(\theta_{\text{dyn}})), \quad \mathcal{L}_4 = \text{KL}(q(\theta_{\text{dec}}) \| p(\theta_{\text{dec}})). \tag{60}$$

### B.2 Computation of ELBO.

We compute the ELBO using the following algorithm:

1. Sample $\theta_{\text{dyn}}, \theta_{\text{dec}}$ from $q_{\psi_{\text{dyn}}}(\theta_{\text{dyn}}), q_{\psi_{\text{dec}}}(\theta_{\text{dec}})$.

2. Sample $s_{1:B}$ by sampling each $s_b^j$ from $q_{\psi_b^j}(s_b^j)$ with $\psi_b^j = h_{\theta_{\text{enc}}}(u[t_{[b]}, x_j])$.

3. Compute $u_{1:M}$ from $s_{1:B}$ as in Equations 14-16.

4. Compute ELBO $\mathcal{L}$ (KL terms are computed in closed form, for expectations we use Monte Carlo integration with one sample).

Sampling is done using reparametrization to allow unbiased gradients w.r.t. the model parameters.

## C  Appendix C

### C.1  Datasets.

**SHALLOW WATER.**  The shallow water equations are a system of partial differential equations (PDEs) that simulate the behavior of water in a shallow basin. These equations are effectively a depth-integrated version of the Navier-Stokes equations, assuming the horizontal length scale is significantly larger than the vertical length scale. Given these assumptions, they provide a model for water dynamics in a basin or similar environment, and are commonly utilized in predicting the propagation of water waves, tides, tsunamis, and coastal currents. The state of the system modeled by these equations consists of the wave height $h(t, x, y)$, velocity in the $x$-direction $u(t, x, y)$ and velocity in the $y$-direction $v(t, x, y)$. Given an initial state $(h_0, u_0, v_0)$, we solve the PDEs on a spatial domain $\Omega$ over time interval $[0, T]$. The shallow water equations are defined as:

$$\frac{\partial h}{\partial t} + \frac{\partial(hu)}{\partial x} + \frac{\partial(hv)}{\partial y} = 0, \tag{61}$$

$$\frac{\partial u}{\partial t} + u\frac{\partial u}{\partial x} + v\frac{\partial u}{\partial y} + g\frac{\partial h}{\partial x} = 0, \tag{62}$$

$$\frac{\partial v}{\partial t} + u\frac{\partial v}{\partial x} + v\frac{\partial v}{\partial y} + g\frac{\partial h}{\partial y} = 0, \tag{63}$$

where $g$ is the gravitational constant.

The spatial domain $\Omega$ is a unit square with periodic boundary conditions. We set $T = 0.1$ sec. The solution is evaluated at randomly selected spatial locations and time points. We use 1089 spatial locations and 25 time points. The spatial end temporal grids are the same for all trajectories. Since we are dealing with partially-observed cases, we assume that we observe only the wave height $h(t, x, y)$.

For each trajectory, we start with zero initial velocities and the initial height $h_0(x, y)$ generated as:

$$\tilde{h}_0(x, y) = \sum_{k,l=-N}^{N} \lambda_{kl} \cos(2\pi(kx + ly)) + \gamma_{kl} \sin(2\pi(kx + ly)), \tag{64}$$

$$h_0(x, y) = 1 + \frac{\tilde{h}_0(x, y) - \min(\tilde{h}_0)}{\max(\tilde{h}_0) - \min(\tilde{h}_0)}, \tag{65}$$

where $N = 3$ and $\lambda_{kl}, \gamma_{kl} \sim \mathcal{N}(0, 1)$.

The datasets used for training, validation, and testing contain 60, 20, and 20 trajectories, respectively.

We use scikit-fdiff (Cellier, 2019) to solve the PDEs.

**NAVIER-STOKES.**  For this dataset we model the propagation of a scalar field (e.g., smoke concentration) in a fluid (e.g., air). The modeling is done by coupling the Navier-Stokes equations with the Boussinesq buoyancy term and the transport equation to model the propagation of the scalar field. The state of the system modeled by these equations consists of the scalar field $c(t, x, y)$, velocity in $x$-direction $u(t, x, y)$, velocity in $y$-direction $v(t, x, y)$, and pressure $p(t, x, y)$. Given an initial state

$(c_0, u_0, v_0, p_0)$, we solve the PDEs on a spatial domain $\Omega$ over time interval $[0, T]$. The Navier-Stokes equations with the transport equation are defined as:

$$\frac{\partial u}{\partial x} + \frac{\partial v}{\partial y} = 0, \tag{66}$$

$$\frac{\partial u}{\partial t} + u\frac{\partial u}{\partial x} + v\frac{\partial u}{\partial y} = -\frac{\partial p}{\partial x} + \nu\left(\frac{\partial^2 u}{\partial x^2} + \frac{\partial^2 u}{\partial y^2}\right), \tag{67}$$

$$\frac{\partial v}{\partial t} + u\frac{\partial v}{\partial x} + v\frac{\partial v}{\partial y} = -\frac{\partial p}{\partial y} + \nu\left(\frac{\partial^2 v}{\partial x^2} + \frac{\partial^2 v}{\partial y^2}\right) + c, \tag{68}$$

$$\frac{\partial c}{\partial t} = -u\frac{\partial c}{\partial x} - v\frac{\partial c}{\partial y} + \nu\left(\frac{\partial^2 c}{\partial x^2} + \frac{\partial^2 c}{\partial y^2}\right), \tag{69}$$

where $\nu = 0.002$.

The spatial domain $\Omega$ is a unit square with periodic boundary conditions. We set $T = 2.0$ sec, but drop the first $0.5$ seconds due to slow dynamics during this time period. The solution is evaluated at randomly selected spatial locations and time points. We use 1089 spatial locations and 25 time points. The spatial and temporal grids are the same for all trajectories. Since we are dealing with partially-observed cases, we assume that we observe only the scalar field $c(t, x, y)$.

For each trajectory, we start with zero initial velocities and pressure, and the initial scalar field $c_0(x, y)$ is generated as:

$$\tilde{c}_0(x, y) = \sum_{k,l=-N}^{N} \lambda_{kl} \cos(2\pi(kx + ly)) + \gamma_{kl} \sin(2\pi(kx + ly)), \tag{70}$$

$$c_0(x, y) = \frac{\tilde{c}_0(x, y) - \min(\tilde{c}_0)}{\max(\tilde{c}_0) - \min(\tilde{c}_0)}, \tag{71}$$

where $N = 2$ and $\lambda_{kl}, \gamma_{kl} \sim \mathcal{N}(0, 1)$.

The datasets used for training, validation, and testing contain 60, 20, and 20 trajectories, respectively.

We use PhiFlow (Holl et al., 2020) to solve the PDEs.

**SCALAR FLOW.** This dataset, proposed by Eckert et al. (2019), consists of observations of smoke plumes rising in hot air. The observations are post-processed camera images of the smoke plumes taken from multiple views. For simplicity, we use only the front view. The dataset contains 104 trajectories, where each trajectory has 150 time points and each image has the resolution $1080 \times 1920$. Each trajectory was recorded for $T = 2.5$ seconds.

To reduce dimensionality of the observations we sub-sample the original spatial and temporal grids. For the temporal grid, we remove the first 50 time points, which leaves 100 time points, and then take every 4th time point, thus leaving 20 time points in total. The original $1080 \times 1920$ spatial grid is first down-sampled by a factor of 9 giving a new grid with resolution $120 \times 213$, and then the new grid is further sub-sampled based on the smoke density at each node. In particular, we compute the average smoke density at each node (averaged over time), and then sample the nodes without replacement with the probability proportional to the average smoke density (thus, nodes that have zero density most of the time are not selected). See example of a final grid in Figure 9. This gives a new grid with 1089 nodes.

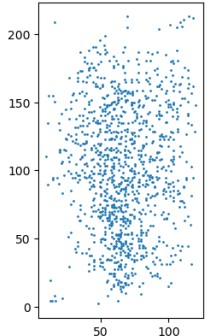

Figure 9: Spatial grid used for SCALAR FLOW dataset.

We further smooth the observations by applying Gaussian smoothing with the standard deviation of 1.5 (assuming domain size $120 \times 213$).

We use the first 60 trajectories for training, next 20 for validation and next 20 for testing.

In this case the spatial domain is non-periodic, which means that for some observation location $\boldsymbol{x}_i$ some of its spatial neighbors $\mathcal{N}_S(\boldsymbol{x}_i)$ might be outside of the domain. We found that to account for such cases it is sufficient to mark such out-of-domain neighbors by setting their value to $-1$.

Time grids used for the three datasets are shown in Figure 10.

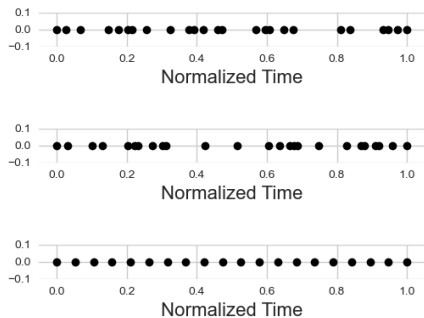

Figure 10: Time grids used for SHALLOW WATER (top), NAVIER-STOKES (middle), and SCALAR FLOW (bottom).

## C.2 Model architecture and hyper-parameters.

**Dynamics function.** For all datasets we define $F_{\theta_{\text{dyn}}}$ as an MLP. For SHALLOW WATER/NAVIER-STOKES/SCALAR FLOW we use 1/3/3 hidden layers with the size of 1024/512/512, respectively. We use ReLU nonlinearities.

**Observation function.** For all datasets we define $g_{\theta_{\text{dec}}}$ as a selector function which takes the latent state $\boldsymbol{z}(t, x) \in \mathbb{R}^d$ and returns its first component.

**Encoder.** Our encoder $h_{\theta_{\text{enc}}}$ consists of three function: $h_{\theta_{\text{spatial}}}$, $h_{\theta_{\text{temporal}}}$, and $h_{\theta_{\text{read}}}$. The spatial aggregation function $h_{\theta_{\text{spatial}}}$ is a linear mapping to $\mathbb{R}^{128}$. The temporal aggregation function $h_{\theta_{\text{temporal}}}$ is a stack of transformer layers with temporal attention and continuous relative positional encodings (Iakovlev et al., 2023). For all datasets, we set the number of transformer layers to 6. Finally, the variational parameter readout function $h_{\theta_{\text{read}}}$ is a mapping defined as

$$\boldsymbol{\psi}_b^j = h_{\theta_{\text{read}}}(\alpha_{[b]}^{\text{T}}) = \begin{pmatrix} \boldsymbol{\gamma}_b^j \\ \boldsymbol{\tau}_b^j \end{pmatrix} = \begin{pmatrix} \text{Linear}(\alpha_{[b]}^{\text{T}}) \\ \exp(\text{Linear}(\alpha_{[b]}^{\text{T}})) \end{pmatrix}, \tag{72}$$

where Linear is a linear layer (different for each line), and $\boldsymbol{\gamma}_b^j$ and $\boldsymbol{\tau}_b^j$ are the variational parameters discussed in Appendix A.

**Spatial and temporal neighborhoods.** We use the same spatial neighborhoods $\mathcal{N}_{\text{S}}(\boldsymbol{x})$ for both the encoder and the dynamics function. We define $\mathcal{N}_{\text{S}}(\boldsymbol{x})$ as the set of points consisting of the point $\boldsymbol{x}$ and points on two concentric circles centered at $\boldsymbol{x}$, with radii $r$ and $r/2$, respectively. Each circle contains 8 points spaced 45 degrees apart (see Figure 11 (right)). The radius $r$ is set to 0.1. For SHALLOW WATER/NAVIER-STOKES/SCALAR FLOW the size of temporal neighborhood ($\delta_T$) is set to 0.1/0.1/0.2, respectively.

**Multiple Shooting.** For SHALLOW WATER/NAVIER-STOKES/SCALAR FLOW we split the full training trajectories into 4/4/19 sub-trajectories, or, equivalently, have the sub-trajectory length of 6/6/2.

## C.3 Training, validation, and testing setup.

**Data preprocessing.** We scale the temporal grids, spatial grids, and observations to be within the interval $[0, 1]$.

**Training.** We train our model for 20000 iterations using Adam (Kingma and Ba, 2017) optimizer with constant learning rate 3e-4 and linear warmup for 200 iterations. The latent spatiotemporal dynamics are simulated using differentiable ODE solvers from the torchdiffeq package (Chen, 2018) (we use dopri5 with rtol=1e-3, atol=1e-4, no adjoint). The batch size is 1.

**Validation.** We use validation set to track the performance of our model during training and save the parameters that produce the best validation performance. As performance measure we use the mean absolute error at predicting the full validation trajectories given some number of initial observations. For SHALLOW WATER/NAVIER-STOKES/SCALAR FLOW we use the first 5/5/10 observations. The predictions are made by taking one sample from the posterior predictive distribution (see Appendix C.4 for details).

**Testing.** Testing is done similarly to validation, except that as the prediction we use an estimate of the expected value of the posterior predictive distribution (see Appendix C.4 for details).

### C.4 Forecasting.

Given initial observations $\tilde{\boldsymbol{u}}_{1:m}$ at time points $t_{1:m}$, we predict the future observation $\tilde{\boldsymbol{u}}_n$ at a time point $t_n > t_m$ as the expected value of the approximate posterior predictive distribution:

$$p(\tilde{\boldsymbol{u}}_n|\tilde{\boldsymbol{u}}_{1:m}, \boldsymbol{u}_{1:M}) \approx \int p(\tilde{\boldsymbol{u}}_n|\tilde{\boldsymbol{s}}_m, \theta_{\mathrm{dyn}}, \theta_{\mathrm{dec}})q(\tilde{\boldsymbol{s}}_m)q(\theta_{\mathrm{dyn}})q(\theta_{\mathrm{dec}})d\tilde{\boldsymbol{s}}_m d\theta_{\mathrm{dyn}} d\theta_{\mathrm{dec}}. \tag{73}$$

The expected value is estimated via Monte Carlo integration, so the algorithm for predicting $\tilde{\boldsymbol{u}}_n$ is:

1. Sample $\theta_{\mathrm{dyn}}, \theta_{\mathrm{dec}}$ from $q(\theta_{\mathrm{dyn}}), q(\theta_{\mathrm{dec}})$.
2. Sample $\tilde{\boldsymbol{s}}_m$ from $q(\tilde{\boldsymbol{s}}_m) = \prod_{j=1}^{N} q_{\boldsymbol{\psi}_m^j}(\tilde{\boldsymbol{s}}_m^j)$, where the variational parameters $\boldsymbol{\psi}_m^j$ are given by the encoder $h_{\theta_{\mathrm{enc}}}$ operating on the initial observations $\tilde{\boldsymbol{u}}_{1:m}$ as $\boldsymbol{\psi}_m^j = h_{\theta_{\mathrm{enc}}}(\tilde{\boldsymbol{u}}[t_m, \boldsymbol{x}_j])$.
3. Compute the latent state $\tilde{\boldsymbol{z}}(t_n) = \boldsymbol{z}(t_n; t_m, \tilde{\boldsymbol{s}}_m, \theta_{\mathrm{dyn}})$.
4. Sample $\tilde{\boldsymbol{u}}_n$ by sampling each $\tilde{\boldsymbol{u}}_n^j$ from $\mathcal{N}(\tilde{\boldsymbol{u}}_n^j|g_{\theta_{\mathrm{dec}}}(\tilde{\boldsymbol{z}}(t_n, \boldsymbol{x}_j))), \sigma_u^2 I)$.
5. Repeat steps 1-4 $n$ times and average the predictions (we use $n = 10$).

### C.5 Model comparison setup.

**NODE.** For the NODE model the dynamics function was implemented as a fully connected feedforward neural network with 3 hidden layers, 512 neurons each, and ReLU nonlinearities.

**FEN.** We use the official implementation of FEN. We use FEN variant without the transport term as we found it improves results on our datasets. The dynamics were assumed to be stationary and autonomous in all cases. The dynamics function was represented by a fully connected feedforward neural network with 3 hidden layers, 512 neurons each, and ReLU nonlinearities.

**NSPDE.** We use the official implementation of NSPDE. We set the number of hidden channels to 16, and set modes1 and modes2 to 32.

**DINo.** We use the official implementation of DINo. The encoder is an MLP with 3 hidden layers, 512 neurons each, and Swish non-linearities. The code dimension is 100. The dynamics function is an MLP with 3 hidden layers, 512 neurons each, and Swish non-linearities. The decoder has 3 layers and 64 channels.

**MAgNet.** We use the official implementation of MAgNet. We use the graph neural network variant of the model. The number of message-passing steps is 5. All MLPs have 4 layers with 128 neurons each in each layer. The latent state dimension is 128.

## D  Appendix D

### D.1  Spatiotemporal neighborhood shapes and sizes.

Here we investigate the effect of changing the shape and size of spatial and temporal neighborhoods used by the encoder and dynamics functions. We use the default hyperparameters discussed in Appendix C and change only the neighborhood shape or size. A neighborhood size of zero implies no spatial/temporal aggregation.

Initially, we use the original circular neighborhood displayed in Figure 11 for both encoder and dynamics function and change only its size (radius). The results are presented in Figures 12a and 12b. In Figure 12a, it is surprising to see very little effect from changing the encoder's spatial neighborhood size. A potential explanation is that the dynamics function shares the spatial aggregation task with the encoder. However, the results in Figure 12b are more intuitive, displaying a U-shaped curve for the test MAE, indicating the importance of using spatial neighborhoods of appropriate size. Interestingly, the best results tend to be achieved with relatively large neighborhood sizes. Similarly, Figure 12c shows U-shaped curves for the encoder's temporal neighborhood size, suggesting that latent state inference benefits from utilizing local temporal information.

We then examine the effect of changing the shape of the dynamics function's spatial neighborhood. We use $n$circle neighborhoods, which consist of $n$ equidistant concentric circular neighborhoods (see examples in Figure 11). Effectively, we maintain a fixed neighborhood size while altering its density. The results can be seen in Figure 13. We find that performance does not significantly improve when using denser (and presumably more informative) spatial neighborhoods, indicating that accurate predictions only require a relatively sparse neighborhood with appropriate size.

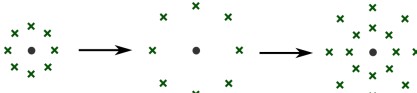

Figure 11: **Left:** original circular neighborhood (1circle). **Center:** circular neighborhood with increased size. **Right:** circular neighborhood of a different shape (2circle).

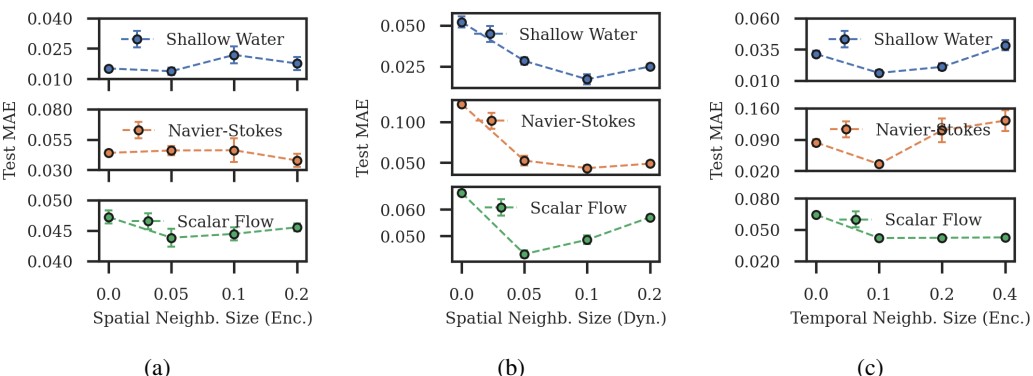

Figure 12: **(a),(b):** Test MAE vs spatial neighborhood sizes of the encoder and dynamics function, respectively. **(c):** Test MAE vs temporal neighborhood size of the encoder. Note that the spatial and temporal domains are normalized, so their largest size in any dimension is 1.

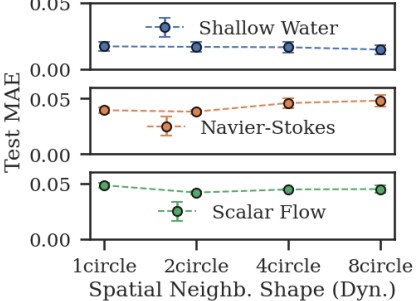

Figure 13: Test MAE vs spatial neighborhood shape.

### D.2 Multiple shooting.

Here we demonstrate the effect of using multiple shooting for model training. In Figure 14 (left), we vary the sub-trajectory length (longer sub-trajectories imply more difficult training) and plot the test errors for each sub-trajectory length. We observe that in all cases, the best results are achieved when the sub-trajectory length is considerably smaller than the full trajectory length. In Figure 14 (right) we further show the training times, and as can be seen multiple shooting allows to noticeably reduce the training times.

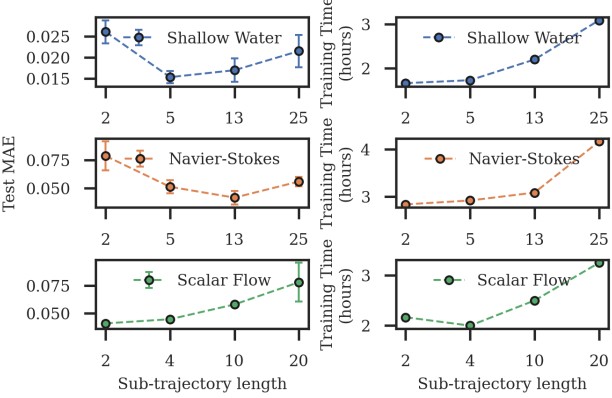

Figure 14: Test MAE vs training sub-trajectory length.

## E   Appendix E

**Noisy Data.**   Here we show the effect of observation noise on our model and compare the results against other models. We train all models with data noise of various strengths, and then compute test MAE on noiseless data (we still use noisy data to infer the initial state at test time). Figure 15 shows that our model can manage noise strength up to 0.1 without significant drops in performance. Note that all observations are in the range $[0, 1]$.

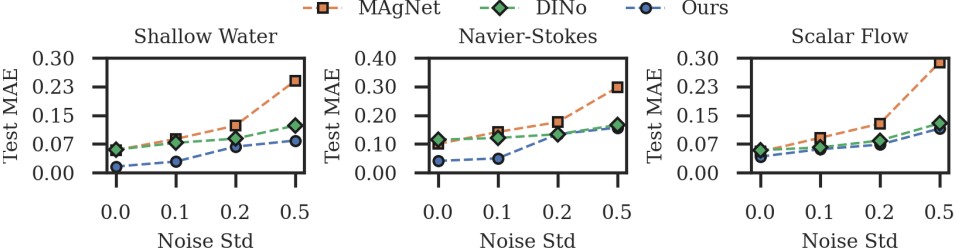

Figure 15: Test MAE vs observation noise $\sigma_u$.

## F   Appendix F

### F.1   Model Predictions

We show (Fig. 16) predictions of different models trained on different datasets (synthetic data is partially observed).

### F.2   Visualization of Prediction Uncertainty

Figures 17, 18, and 19 demonstrate the prediction uncertainty across different samples from the posterior distribution.

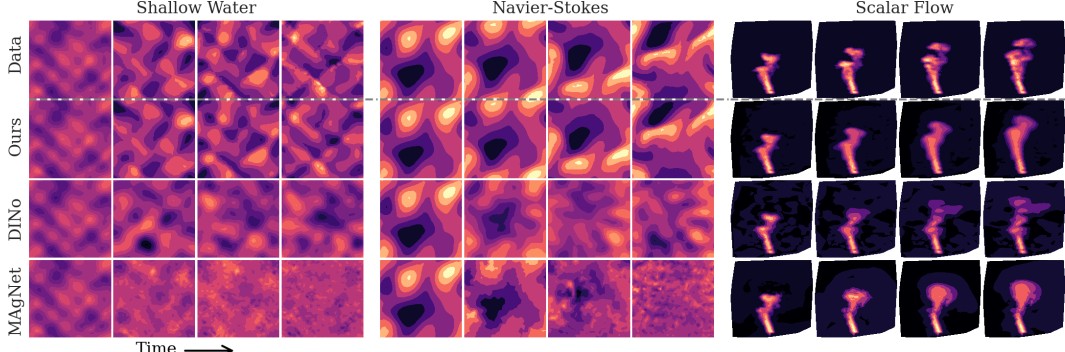

Figure 16: Predictions from different models. Only forecasting is shown.

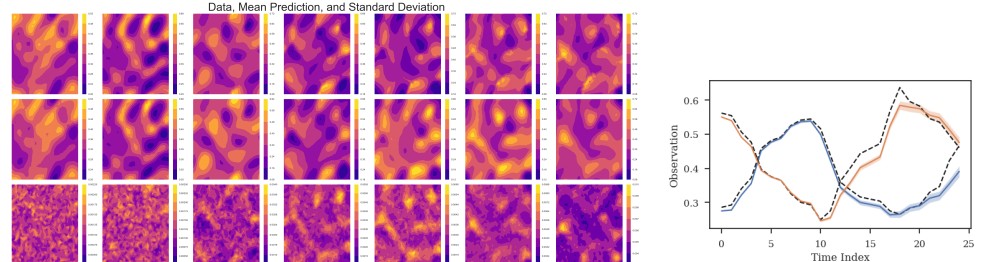

Figure 17: **Left:** Test prediction for a single trajectory on SHALLOW WATER dataset. Show are data (top), mean prediction (middle), and standard deviation of the predictions (bottom). Columns show predictions at consecutive time points. **Right:** Ground truth observations (dashed black) and predictions (colored) with standard deviation plotted over time at two spatial locations.

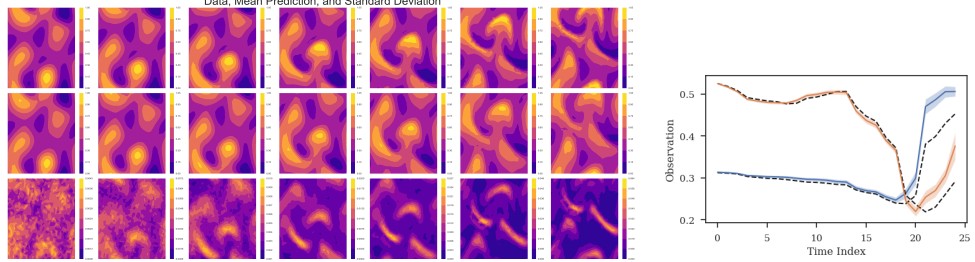

Figure 18: **Left:** Test prediction for a single trajectory on NAVIER-STOKES dataset. Show are data (top), mean prediction (middle), and standard deviation of the predictions (bottom). Columns show predictions at consecutive time points. **Right:** Ground truth observations (dashed black) and predictions (colored) with standard deviation plotted over time at two spatial locations.

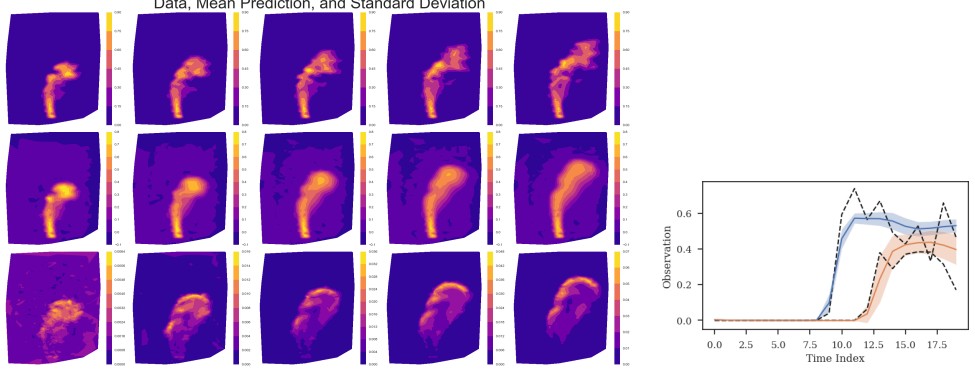

Figure 19: **Left:** Test prediction for a single trajectory on SCALAR FLOW dataset. Show are data (top), mean prediction (middle), and standard deviation of the predictions (bottom). Columns show predictions at consecutive time points. **Right:** Ground truth observations (dashed black) and predictions (colored) with standard deviation plotted over time at two spatial locations.

