## A  Appendix A

### A.1  Model specification.

Here we provide all details about our model specification. The joint distribution for our model is

$$p(\boldsymbol{u}_{1:M}, \boldsymbol{s}_{1:B}, \theta_{\text{dyn}}, \theta_{\text{dec}}) = p(\boldsymbol{u}_{1:N}|\boldsymbol{s}_{1:B}, \theta_{\text{dyn}}, \theta_{\text{dec}})p(\boldsymbol{s}_{1:B}|\theta_{\text{dyn}})p(\theta_{\text{dyn}})p(\theta_{\text{dec}}). \tag{23}$$

Next, we specify each component in detail.

**Parameter priors.**  The parameter priors are isotropic zero-mean multivariate normal distributions:

$$p(\theta_{\text{dyn}}) = \mathcal{N}(\theta_{\text{dyn}}|\mathbf{0}, I), \tag{24}$$

$$p(\theta_{\text{dec}}) = \mathcal{N}(\theta_{\text{dec}}|\mathbf{0}, I), \tag{25}$$

where $\mathcal{N}$ is the normal distribution, $\mathbf{0}$ is a zero vector, and $I$ is the identity matrix, both have an appropriate dimensionality dependent on the number of encoder and dynamics parameters.

**Continuity prior.**  We define the continuity prior as

$$p(\boldsymbol{s}_{1:B}|\theta_{\text{dyn}}) = p(\boldsymbol{s}_1) \prod_{b=2}^{B} p(\boldsymbol{s}_b|\boldsymbol{s}_{b-1}, \theta_{\text{dyn}}), \tag{26}$$

$$= \left[ \prod_{j=1}^{N} p(\boldsymbol{s}_1^j) \right] \left[ \prod_{b=2}^{B} \prod_{j=1}^{N} p(\boldsymbol{s}_b^j|\boldsymbol{s}_{b-1}, \theta_{\text{dyn}}) \right], \tag{27}$$

$$= \left[ \prod_{j=1}^{N} \mathcal{N}(\boldsymbol{s}_1^j|\mathbf{0}, I) \right] \left[ \prod_{b=2}^{B} \prod_{j=1}^{N} \mathcal{N}\left( \boldsymbol{s}_b^j|\boldsymbol{z}(t_{[b]}, \boldsymbol{x}_j; t_{[b-1]}, \boldsymbol{s}_{b-1}, \theta_{\text{dyn}}), \sigma_c^2 I \right) . \right], \tag{28}$$

where $\mathcal{N}$ is the normal distribution, $\mathbf{0} \in \mathbb{R}^d$ is a zero vector, $I \in \mathbb{R}^{d \times d}$ is the identity matrix, and $\sigma_c \in \mathbb{R}$ is the parameter controlling the strength of the prior. Smaller values of $\sigma_c$ tend to produce smaller gaps between the sub-trajectories.

**Observation model**

$$p(\boldsymbol{u}_{1:N}|\boldsymbol{s}_{1:B}, \theta_{\text{dyn}}, \theta_{\text{dec}}) = \prod_{b=1}^{B} \prod_{i \in \mathcal{I}_b} \prod_{j=1}^{N} p(\boldsymbol{u}_i^j|\boldsymbol{s}_b, \theta_{\text{dyn}}, \theta_{\text{dec}}) \tag{29}$$

$$= \prod_{b=1}^{B} \prod_{i \in \mathcal{I}_b} \prod_{j=1}^{N} p(\boldsymbol{u}_i^j|g_{\theta_{\text{dec}}}(\boldsymbol{z}(t_i, \boldsymbol{x}_j; t_{[b]}, \boldsymbol{s}_b, \theta_{\text{dyn}}))) \tag{30}$$

$$= \prod_{b=1}^{B} \prod_{i \in \mathcal{I}_b} \prod_{j=1}^{N} \mathcal{N}(\boldsymbol{u}_i^j|g_{\theta_{\text{dec}}}(\boldsymbol{z}(t_i, \boldsymbol{x}_j; t_{[b]}, \boldsymbol{s}_b, \theta_{\text{dyn}})), \sigma_u^2 I), \tag{31}$$

where $\mathcal{N}$ is the normal distribution, $\sigma_u^2$ is the observation noise variance, and $I \in \mathbb{R}^{D \times D}$ is the identity matrix. Note again that $\boldsymbol{z}(t_i, \boldsymbol{x}_j; t_{[b]}, \boldsymbol{s}_b, \theta_{\text{dyn}})$ above equals the ODE forward solution ODESolve$(t_i; t_{[b]}, \boldsymbol{s}_b, \theta_{\text{dyn}})$ at grid location $\boldsymbol{x}_j$.

### A.2  Approximate posterior specification.

Here we provide all details about the approximate posterior. We define the approximate posterior as

$$q(\theta_{\text{dyn}}, \theta_{\text{dec}}, \boldsymbol{s}_{1:B}) = q(\theta_{\text{dyn}})q(\theta_{\text{dec}})q(\boldsymbol{s}_{1:B}) = q_{\psi_{\text{dyn}}}(\theta_{\text{dyn}})q_{\psi_{\text{dec}}}(\theta_{\text{dec}}) \prod_{b=1}^{B} \prod_{j=1}^{N} q_{\psi_b^j}(\boldsymbol{s}_b^j). \tag{32}$$

Next, we specify each component in detail.

**Dynamics parameters posterior.** We define $q_{\psi_{\mathrm{dyn}}}(\theta_{\mathrm{dyn}})$ as

$$q_{\psi_{\mathrm{dyn}}}(\theta_{\mathrm{dyn}}) = \mathcal{N}(\theta_{\mathrm{dyn}}|\gamma_{\mathrm{dyn}}, \mathrm{diag}(\tau_{\mathrm{dyn}}^2)), \tag{33}$$

where $\gamma_{\mathrm{dyn}}$ and $\tau_{\mathrm{dyn}}^2$ are vectors with an appropriate dimension (dependent on the number of dynamics parameters), and $\mathrm{diag}(\tau_{\mathrm{dyn}}^2)$ is a matrix with $\tau_{\mathrm{dyn}}^2$ on the diagonal. We define the vector of variational parameters as $\psi_{\mathrm{dyn}} = (\gamma_{\mathrm{dyn}}, \tau_{\mathrm{dyn}}^2)$. We optimize directly over $\psi_{\mathrm{dyn}}$ and initialize $\gamma_{\mathrm{dyn}}$ using Xavier (Glorot and Bengio, 2010) initialization, while $\tau_{\mathrm{dyn}}$ is initialized with each element equal to $9 \cdot 10^{-4}$.

**Decoder parameters posterior.** We define $q_{\psi_{\mathrm{dec}}}(\theta_{\mathrm{dec}})$ as

$$q_{\psi_{\mathrm{dec}}}(\theta_{\mathrm{dec}}) = \mathcal{N}(\theta_{\mathrm{dec}}|\gamma_{\mathrm{dec}}, \mathrm{diag}(\tau_{\mathrm{dec}}^2)), \tag{34}$$

where $\gamma_{\mathrm{dec}}$ and $\tau_{\mathrm{dec}}^2$ are vectors with an appropriate dimension (dependent on the number of decoder parameters), and $\mathrm{diag}(\tau_{\mathrm{dec}}^2)$ is a matrix with $\tau_{\mathrm{dec}}^2$ on the diagonal. We define the vector of variational parameters as $\psi_{\mathrm{dec}} = (\gamma_{\mathrm{dec}}, \tau_{\mathrm{dec}}^2)$. We optimize directly over $\psi_{\mathrm{dec}}$ and initialize $\gamma_{\mathrm{dec}}$ using Xavier (Glorot and Bengio, 2010) initialization, while $\tau_{\mathrm{dec}}$ is initialized with each element equal to $9 \cdot 10^{-4}$.

**Shooting variables posterior.** We define $q_{\psi_b^j}(s_b^j)$ as

$$q_{\psi_b^j}(s_b^j) = \mathcal{N}(s_b^j|\gamma_b^j, \mathrm{diag}([\tau_b^j]^2))), \tag{35}$$

where the vectors $\gamma_b^j, \tau_b^j \in \mathbb{R}^d$ are returned by the encoder $h_{\theta_{\mathrm{enc}}}$, and $\mathrm{diag}([\tau_b^j]^2)$ is a matrix with $[\tau_b^j]^2$ on the diagonal. We define the vector of variational parameters as $\psi_b^j = (\gamma_b^j, [\tau_b^j])$. Because the variational inference for the shooting variables is amortized, our model is trained w.r.t. the parameters of the encoder network, $\theta_{\mathrm{enc}}$.

# B  Appendix B

## B.1  Derivation of ELBO.

For our model and the choice of the approximate posterior the ELBO can be written as

$$\mathcal{L} = \int q(\theta_{\mathrm{dyn}}, \theta_{\mathrm{dec}}, s_{1:B}) \ln \frac{p(u_{1:M}, s_{1:B}, \theta_{\mathrm{dyn}}, \theta_{\mathrm{dec}})}{q(\theta_{\mathrm{dyn}}, \theta_{\mathrm{dec}}, s_{1:B})} d\theta_{\mathrm{dyn}} d\theta_{\mathrm{dec}} ds_{1:B} \tag{36}$$

$$= \int q(\theta_{\mathrm{dyn}}, \theta_{\mathrm{dec}}, s_{1:B}) \ln \frac{p(u_{1:M}|s_{1:B}, \theta_{\mathrm{dyn}}, \theta_{\mathrm{dec}})p(s_{1:B}|\theta_{\mathrm{dyn}})p(\theta_{\mathrm{dyn}})p(\theta_{\mathrm{dec}})}{q(s_{1:B})q(\theta_{\mathrm{dyn}})q(\theta_{\mathrm{dec}})} d\theta_{\mathrm{dyn}} d\theta_{\mathrm{dec}} ds_{1:B} \tag{37}$$

$$= \int q(\theta_{\mathrm{dyn}}, \theta_{\mathrm{dec}}, s_{1:B}) \ln p(u_{1:M}|s_{1:B}, \theta_{\mathrm{dyn}}, \theta_{\mathrm{dec}}) d\theta_{\mathrm{dyn}} d\theta_{\mathrm{dec}} ds_{1:B} \tag{38}$$

$$- \int q(\theta_{\mathrm{dyn}}, \theta_{\mathrm{dec}}, s_{1:B}) \ln \frac{q(s_{1:B})}{p(s_{1:B}|\theta_{\mathrm{dyn}})} d\theta_{\mathrm{dyn}} d\theta_{\mathrm{dec}} ds_{1:B} \tag{39}$$

$$- \int q(\theta_{\mathrm{dyn}}, \theta_{\mathrm{dec}}, s_{1:B}) \ln \frac{q(\theta_{\mathrm{dyn}})}{p(\theta_{\mathrm{dyn}})} d\theta_{\mathrm{dyn}} d\theta_{\mathrm{dec}} ds_{1:B} \tag{40}$$

$$- \int q(\theta_{\mathrm{dec}}, \theta_{\mathrm{dec}}, s_{1:B}) \ln \frac{q(\theta_{\mathrm{dec}})}{p(\theta_{\mathrm{dec}})}

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

487 We set the spatial domain $\Omega$ to be a unit square and use periodic boundary conditions. We set $T = 0.1$.
488 The solution is evaluated at randomly selected spatial locations and time points. We use $1089$ spatial
489 locations and $25$ time points. The spatial end temporal grids are the same for all trajectories. Since we
490 are dealing with partially-observed cases, we assume that we observe only the wave height $h(t, x, y)$.

491 For each trajectory, we start with zero initial velocities and the initial height $h_0(x, y)$ generated as:

$$\tilde{h}_0(x, y) = \sum_{k,l=-N}^{N} \lambda_{kl} \cos(2\pi(kx + ly)) + \gamma_{kl} \sin(2\pi(kx + ly)), \tag{63}$$

$$h_0(x, y) = 1 + \frac{\tilde{h}_0(x, y) - \min(\tilde{h}_0)}{\max(\tilde{h}_0) - \min(\tilde{h}_0)}, \tag{64}$$

492 where $N = 3$ and $\lambda_{kl}, \gamma_{kl} \sim \mathcal{N}(0, 1)$.

493 The datasets used for training, validation, and testing contain 60, 20, and 20 trajectories, respectively.

494 We use scikit-fdiff (Cellier, 2019) to solve the PDEs.

495 **NAVIER-STOKES.** For this dataset we model the propagation of a scalar field (e.g., smoke concen-
496 tration) in a fluid (e.g., air). The modeling is done by coupling the Navier-Stokes equations with the
497 Boussinesq buoyancy term and the transport equation to model the propagation of the scalar field.
498 The state of the system modeled by these equations consists of the scalar field $c(t, x, y)$, velocity in
499 $x$-direction $u(t, x, y)$, velocity in $y$-direction $v(t, x, y)$, and pressure $p(t, x, y)$. Given an initial state

$(c_0, u_0, v_0, p_0)$, we solve the PDEs on a spatial domain $\Omega$ over time interval $[0, T]$. The Navier-Stokes equations with the transport equation are defined as:

$$\frac{\partial u}{\partial x} + \frac{\partial v}{\partial y} = 0, \tag{65}$$

$$\frac{\partial u}{\partial t} + u\frac{\partial u}{\partial x} + v\frac{\partial u}{\partial y} = -\frac{\partial p}{\partial x} + \nu\left(\frac{\partial^2 u}{\partial x^2} + \frac{\partial^2 u}{\partial y^2}\right), \tag{66}$$

$$\frac{\partial v}{\partial t} + u\frac{\partial v}{\partial x} + v\frac{\partial v}{\partial y} = -\frac{\partial p}{\partial y} + \nu\left(\frac{\partial^2 v}{\partial x^2} + \frac{\partial^2 v}{\partial y^2}\right) + c, \tag{67}$$

$$\frac{\partial c}{\partial t} = -u\frac{\partial c}{\partial x} - v\frac{\partial c}{\partial y} + \nu\left(\frac{\partial^2 c}{\partial x^2} + \frac{\partial^2 c}{\partial y^2}\right), \tag{68}$$

where $\nu = 0.002$.

We set the spatial domain $\Omega$ to be a unit square and use periodic boundary conditions. We set $T = 2.0$,

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

{s}_m$ from $q(\tilde{s}_m) = \prod_{j=1}^{N} q_{\psi_m^j}(\tilde{s}_m^j)$, where the variational parameters $\psi_m^j$ are given by the encoder $h_{\theta_{\mathrm{enc}}}$ operating on the initial observations $\tilde{u}_{1:m}$ as $\psi_m^j = h_{\theta_{\mathrm{enc}}}(\tilde{u}[t_m, x_j])$.

3. Compute the latent state $\tilde{z}(t_n) = z(t_n; t_m, \tilde{s}_m, \theta_{\mathrm{dyn}})$.

4. Sample $\tilde{u}_n$ by sampling each $\tilde{u}_n^j$ from $\mathcal{N}(\tilde{u}_n^j | g_{\theta_{\mathrm{dec}}}(\tilde{z}(t_n, x_j))), \sigma_u^2 I)$.

5. Repeat steps 1-4 $n$ times and average the predictions (we use $n = 10$).

### C.5  Model comparison setup.

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

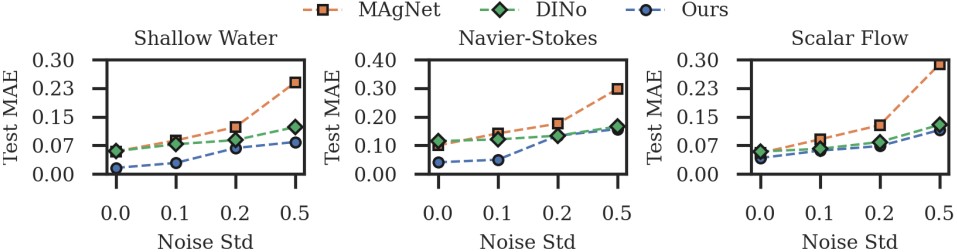

Figure 16: Test MAE vs observation noise $\sigma_u$.