# OpenReview forum: "Learning Space-Time Continuous Latent Neural PDEs from Partially Observed States"
_NeurIPS.cc/2023/Conference — NeurIPS 2023 poster_

### Official Review · Reviewer_nYoy · 2023-06-20

**Soundness:** 3 good
**Presentation:** 4 excellent
**Contribution:** 3 good
**Rating:** 6
**Confidence:** 3

**Summary:**

The authors present a grid-independent generative model to learn PDEs, which supports irregular grids. Amortized variational inference is employed for posterior approximation and multiple shooting, a recent method to train neural ODEs efficiently, is used and adapted to PDEs.
Results show a quite significant dominance over two SOTA PDE learning methods.

**Strengths:**

- The paper is well written and very well presented. The motivation is clear, the model formalism is well presented, the general flow is good. I enjoyed reading the paper.

- The figures are well made and appreciated to have an intuition about the different elements.

- The contribution of the paper is the model itself, which appears to be a natural extension to the Iakovlev et al., 2023, adpated for a PDE instead of an ODE setting. It is therefore not the most original methodologically, but a completely proper and natural contribution.
Note that even though grid-independence and irregular grid prediction capabilities have been studied before for PDE learning, (notably by the SOTA the method is comparing to) which again restrains the novelty of the paper, the present methodology seems to significantly outshine the previous methods, which therefore justifies the contribution.

**Weaknesses:**


- Relating to the previous comment regarding the contribution on the strength section: because it is not a new gap that the method is filling, but rather an improvement over SOTA methods, the contribution of the paper, besides being an extension of the multiple shooting for Neural ODE learning, relies on the performance in order to be strong. Therefore, it would be very useful to have more extensive experiments:
   - of SOTA methods on finer and coarser grids, in order to test super resolution capabilities of the method compared to the other SOTA methods. An experiment is done on Figure 8, but not compared to SOTA methods
   - to test prediction on long future horizon for all models including SOTA
   - trying multiple grid sizes when training, instead of only multiple number of trajectories, as shown in Figure 10

- It seems that it lacks some important details
    -  In the experiments regarding time specification, specifically the time horizon and the time resolution. It is mentioned in the appendix that "25 time points" are used for the two first data and 20 "time points" for the scalar flow, but it is not clear to me what they represent in real time. Maybe I missed it somewhere? If not, please mention it, as this is important when comparing to SOTA methods, as each method seems to have its strength, regarding spatial or time capabilities.

    - In the figures: please add the time resolution on Figure 6 and 9 (how much is each time step?), and the horizon predicted on all the figures, to know what the MAE correspond to! If it is the same every time, please state it clearly somewhere (again, I apologize if I missed it...!)

- How is the generative aspect of the method used? The fact that it is probabilistic is partly what makes it interesting as opposed to SOTA models (MAgNet and DINo, which are not generative), so it is a pity that no probabilistic outputs are used or shown in the results, but rather mean predictions.

I think in general, the authors could move a bit of the formalism of the model to the appendix (which is also taken from Iakovlev et al., 2023 anyway, so that makes it easier), to let some space for more experiments and more details on the experiments, since, once again, they are in my view key to the contribution of the pure performance of the model!

**Questions:**

Please refer to the weaknesses part for recommendations or modifications.

Regarding other small questions or comments:

- A general figure to describe the overall method would be very nice to have! Given, that the method is quite complex and gathers many different elements, it would really help grasping the whole framework, at least at a high level.

- For the spatial aggregation, you use an MLP, like for example in MAgNet. The latter actually shows in the paper, in an ablation study, that the use of the MLP interpolator, is actually not so beneficial with respect to a cubic interpolator. Have you tried other interpolators, perhaps in an ablation study? Since this is the key to the grid-independent capabilities of the model, I think it is worth a part in the paper.

- I think the C.4 appendix section on Forecasting should be put in the main text as it is quite crucial to the method.

- I realized from the appendix training details that the batch size is 1. Was the data so big that it was impossible to fit a higher batch size in memory? Didn't it create training stability issues?

- I think it is a bit unncessary to split the contributions in 3 the way you do it in the introduction : generative model, PDE model, encoder. To my understanding, the model you are proposing is exactly the mix of the 3, so I think it is better to simply do this one strong claim: you designed one PDE learning model, with all its characteristics.

**Limitations:**

The authors addressed some limitations about the scalability of the method, but not much else; perhaps there are other points, like the training speed or the lack of special geometries taken into account within the method, or super-resolution capabilities still to be compared to SOTA? I do not see any potential negative societal impact.

---

> ### Author Rebuttal · Authors · 2023-08-09
>
> Thank you for your detailed review. We address the raised concerns below.
>
> **Additional experiments**
>
> Thank you for your suggestions. We agree that more experiments are required to fully demonstrate the advantages of our method. We conducted multiple additional experiments and present the results below.
>
> First, we extend our comparisons to a larger number of relevant methods. We added three more methods, a simple baseline NeuralODE [1], and two SOTA methods as suggested by the reviewers: Neural Stochastic PDEs (NSPDE) [2], and Finite Element Networks (FEN) [3]. Please see the global rebuttal for description of the setup and results.
>
> Second, to test the performance of our model on predicting over longer time horizons than the ones seen during training, we trained our model only on the first half of the training time grid (i.e., from 0 to 0.5T, where T is the observed time-horizon), and tested it on the full time grid (of length T). The test error for shallow water and scalar flow did not increase, while for Navier-Stokes it increased almost twice. This observation can be understood by the fact that velocity of the species transported in the Navier-Stokes dataset keeps increasing over time. Thus, the system behavior that the model was trained on (before time 0.5T), is different from the system behavior on which the model was tested (after time 0.5T), naturally leading to higher prediction errors. Unfortunately, due to time constraints we were not able to conduct the time horizon experiments for other methods. But we are working on these experiments and will add them to the updated version of our manuscript.
>
> Comparison of the super-resolution performance to other methods was not done due to poor performance of other methods even on the original training grids. However, this is a good suggestion and we will also conduct this experiment and add it to the updated version of our manuscript.
>
> **Time grids**
>
> Indeed, mentioning only the number of time points is not clear enough. For that reason we also provide information about the time horizons (T) in Appendix C.1. For Shallow Water T=0.1sec, for Navier-Stokes T=2sec, while for Scalar Flow T=2.5sec (we did not mention T for scalar flow, but we will add it in the revision). The system evolution over the mentioned time horizons is depicted in Fig. 6. As can be seen, the state of all systems changes a lot during these time intervals. For better intuition, we will visualize the time grids and add the figures into the revised version of our manuscript.
>
> Thank you also for your suggestion for visualizing the forecasting intervals. We will rework the figures to include this suggestion. In our last experiment "Comparison to other models" we discuss the forecasting time intervals, but we agree that this detail should be discussed at the beginning of the Experiments section. We will modify our manuscript accordingly.
>
> **How is the generative aspect of the method used?**
>
> As you correctly noted, we use the mean of the posterior predictive distribution as the prediction. As we discuss in Appendix C.4, the mean is computed by averaging over the predictions corresponding to different samples from the approximate posterior distribution. This is a common way in which uncertainty is incorporated into the model prediction.
>
> **Paper changes**
>
> Thank you for your suggestions about improving our work! We will incorporate them in the revision by moving some of the multiple shooting formalism to the appendix and moving the appendix on forecasting to the main text. We will also work on providing a figure that gives a better overall idea about our method.
>
> **Using MLP for spatial aggregations**
>
> As you correctly noted, we use an MLP as the spatial aggregation function. However, since the spatial aggregation is a part of the encoder, it is used only to infer the latent states, not for interpolation. For interpolation we use standard interpolation methods such as linear interpolation.
>
> In addition to the linear interpolation, we investigated the performance of other interpolation methods: k-NN and inverse distance weighting. Please see the results and description of the experiment in the global rebuttal.
>
> **Using small batch sizes**
>
> Indeed, the batch size was set to 1 for all experiments. The amount of GPU memory used during training was around 4-6GB, so there was a possibility to increase the batch size. However, we did not face any issues with convergence, so we decided to leave it at 1.
>
> **References**
>
> [1] Neural Ordinary Differential Equations, https://arxiv.org/abs/1806.07366
>
> [2] Neural Stochastic PDEs: Resolution-Invariant Learning of Continuous Spatiotemporal Dynamics, https://arxiv.org/abs/2110.10249
>
> [3] Learning the Dynamics of Physical Systems from Sparse Observations with Finite Element Networks, https://openreview.net/forum?id=HFmAukZ-k-2

---

> > ### Comment · Reviewer_nYoy · 2023-08-13
> > **Answer to rebuttal**
> >
> > Thanks for answering to all my comments.
> >
> > I understand all answers and agree with most of them.
> >
> > Just one thing: I am not sure I understand what you mean by your answer about the use of the generative aspect of the method to extract uncertainty outputs. I did understand that you use the mean of the posterior distrib as the prediction, but I was suggesting that you could use the different samples from the posterior to extract a distrib and therefore some uncertainty instead of simply using the mean. Could you elaborate a bit more on why you do not do so?
> >
> > Otherwise, I appreciate your efforts regarding the extra experiments on long horizon and super resolution, on top of the ones for extra models, and I am more than willing to even increase my score to 7 should you perform them in the next revision.
> > If you further improve the structure and the flow of the paper, which is, and I agree with the other reviewers on this, a little dense and hard to read at times, it will be a good contribution to the conference, definitely publishable in my opinion.

---

> > > ### Author Response · Authors · 2023-08-17
> > >
> > > Thank you for your positive comments. We address the raised concerns below.
> > >
> > > Indeed, the way our model makes predictions is similar to what you described. Our proposed method uses variational inference to estimate posteriors of model parameters (both the dynamics model and decoder) and latent variables (initial states). We then take n samples from the posterior (over the initial state and model parameters), generate n predictions, and then average them to obtain the final prediction which we use to compute the test errors and for comparison with other models. In probabilistic modeling literature that corresponds to the standard way of estimating the posterior mean prediction. As you correctly pointed out, the estimated posterior can also be used to compute the (full) predictive density over the test data points and thus allows computing probabilistic test error metrics, such as the expected log predictive density. However, previous methods can only make point predictions and thus we need to resort to posterior mean predictions to make meaningful comparisons. We agree that visualizing the full posterior predictive distributions will be interesting and can provide additional insight. We will include such demonstrations in the next revision.
> > >
> > > We also kindly note that making a revision is possible only after the discussion period, hence all additions (e.g. extra experiments on long horizon and super resolution) and improvements to the flow of the paper will be added after the discussion period is finished.

---

### Official Review · Reviewer_9sfV · 2023-06-29

**Soundness:** 3 good
**Presentation:** 1 poor
**Contribution:** 2 fair
**Rating:** 5
**Confidence:** 4

**Summary:**

The authors proposed a new method in learning PDEs by combining interpolation and NODEs. The difference from NODEs is that the authors uses spatial derivatives in hidden dynamics as PDE, compute the loss between shooting gaps with KL divergence, and compute the initial conditions with a transformer. The authors then support their claim by experimenting with three datasets, shallow water, navier-stokes, and scalar flow.

**Strengths:**

- The authors present a combination of various methods as a new solution to learning PDEs.
- The performance of the method, on the particular datasets the authors choose, are quite well.

**Weaknesses:**

Overall, the proposed method might be nice, but the paper itself is very hard to read.
- In section 3.2 generative modeling, there are two paragraphs discussing multiple shooting. There is no discussion on how multiple shooting is used on generative modeling.
- There are too few captions for the figures. In figure 9, which parts are the time you interpolate? Which parts are extrapolation / forecasting?
- The notations $q_\psi(\theta)$ is so confusing.


**Questions:**

- In fig. 4 minimizing gaps, which norm do you use? Do you instead maximize the probability / likelihood in equation 17? In some commonly used PDEs, such as wave equations, derivatives of the PDE also matters. Did you take them into account? Or does your method only work on specific type of PDEs?
- The experimental results in the paper is very different from the experimental results in DINO paper. Do you use a different experimental setup?
- Are figure 8 and figure 9 partially the same?
- What is a true latent dimension in line 225?
- What are $h$ in figure 5? Are they intemediate states? But in equation 22 they look like functions. Are they the same $h$?
- Also in figure 5, did you add extra observations around $x_j$? Or are they within the dataset? Did you assume all observations are on the same position but not grid-like? Then why don't we use NODEs with finite element meshes?
- What types of problems are these experiments? Why are they interesting? Are they experiments from the SOTA / benchmark you compare with, or are they associated with any practical problems in application?
- For those synthetic experiments, what are the noise level? What is the hidden equation? Navier-stokes might be self evident, but what about shallow water?
- Do the models rely on initialization? How much runs did you run your experiments?

---

> ### Author Rebuttal · Authors · 2023-08-09
>
> Thank you for your detailed review. We address the raised concerns below.
>
> **Using multiple shooting in generative modeling, and gap minimization.**
>
> Indeed, the first two paragraphs of Section 3.2 only motivate the use of multiple shooting, but the rest of the section goes into details about how it is applied in our model. See especially Eq. 14-17 which describe our generative model and the role of multiple shooting in it.
>
> To reduce the gaps we minimize the KL-divergence between the approximate posteriors of the shooting states and continuity prior (see part (iii) of the ELBO, Sec. 4.1). Note that the gap minimization via the KL-divergence is obtained directly from the variational lower bound (see the ELBO derivation in Appendix B.1) because the multiple shooting is formulated as part of our proposed generative model. During minimization we take the whole latent state into account, which means our method is not restricted to specific types of PDEs.
>
> **Do the models rely on initialization? How much runs did you run your experiments?**
>
> As for any other deep-learning-based model, appropriate initialization helps our model to achieve better performance. As we discuss in Appendix A.2, we use the standard Xavier initialization. Also, as we mention in the experiments section, the results are the average over 4 runs with different random seeds. Overall, we observed that the performance of our method is very stable wrt to the random seed changes.
>
> **In Figure 9, which parts are interpolate/forecasting? Are Figures 8 and 9 partially the same?**
>
> In Figure 9 we show only forecasting. We will clarify this point in the revision. Figures 8 and 9 both show predictions of our model, but the results in Figures 8 and 9 are for two different experiments.
>
> **What is a true latent dimension in line 225?**
>
> True latent state dimension refers to the dimension of the full system state, which is 3 for both shallow water and Navier-Stokes equations. For example, for the shallow water equation the state consists of the wave height (scalar) and velocity vector (2-dimensional vector). We will also clarify this point in the revision.
>
> **What are h in figure 5?**
>
> As we discuss at the end of Section 4.2, h are functions that are parts of our encoder for amortized variational inference.
>
> **In figure 5, did you add extra observations around xj?**
>
> No, we do not add any extra observations. We assume that we have observations only at the observation locations. As we discuss in Sec. 4.2, we use interpolation to obtain the system state outside of the observation locations (i.e., the locations marked with crosses around xj in figure 5). This is the core idea that is used both in the encoder and dynamics function which enables to make our proposed model continuous in space and grid-independent. We will clarify this aspect in the revised manuscript.
>
> **What types of problems are these experiments? Why are they interesting? Are they associated with any practical problems in application?**
>
> As we discuss in Sections 1, 2, and 5, our experimental setup is highly-relevant for real-world applications, especially where the observations might be collected over irregular spatiotemporal grid, and the observed states might be incomplete. As our results (and extra comparisons show in the global rebuttal) demonstrate, this is a setting where currently available models fail, while our model demonstrates strong performance highlighting its utility in such challenging scenarios.
>
> **For those synthetic experiments, what are the noise level? What is the hidden equation?**
>
> Data used for experiments in the main section is noiseless. However, in Appendix E we provide comprehensive results for noisy data with different levels of noise. As can be seen, our model is robust and maintains strong performance even under significant noise levels. Also, we provide all details about the systems used for our experiments in Appendix C, including the hidden equations. We will add this detail to the main text.
>
> **The experimental results in the paper is very different from the experimental results in DINO paper. Do you use a different experimental setup?**
>
> Our setup is more challenging than the one used in the DINo paper. They use fully-observed states and 512 training trajectories. In contrast, we use partially observed states and only 60 trajectories. To ensure fair comparisons, we conducted extensive hyperparameter tuning for DINo (as well as for all other baselines comparisons) using the default model parameters in the DINo paper as the starting point.

---

> > ### Comment · Reviewer_9sfV · 2023-08-14
> >
> > The authors provide reasonable explanation on most of my concerns. I am increasing my score accordingly.

---

> > > ### Author Response · Authors · 2023-08-17
> > >
> > > Thank you for your positive comment.

---

### Official Review · Reviewer_mL6m · 2023-07-01

**Soundness:** 2 fair
**Presentation:** 2 fair
**Contribution:** 2 fair
**Rating:** 5
**Confidence:** 4

**Summary:**

This work proposes a latent variable model for PDEs with an encoder-decoder architecture that evolves the latent variables in time with a neural ODE. The model achieves independence of the locations of the evaluation points by linearly interpolating the data onto points distributed in a fixed pattern around each evaluation point.

**Strengths:**

- The paper's main contribution is the grid-independence of the model by interpolating observations and latent states onto fixed neighboring points around each observation point
- The experimental section investigates the effect of the latent state dimension and the size of the training dataset

**Weaknesses:**

- The collocation method is a function space method that finds solutions that solve the PDE at a given set of collocation points. I don't see how this is used in this paper. The authors refer to the general form of first-order-in-time PDE in Eq (4) as the collocation method, but Eq (4) is just a general form of a PDE and not a method.
- The method of lines is only about eliminating spatial derivatives by discretizing the solution for each $t$ in a function space and does not contain any notion of data-driven models or being restricted to evaluations of $z(t, x)$ as claimed in lines 88-89
- Generative modeling of PDE solutions (section 3.2) is neither motivated nor evaluated in experiments
- Use of bayesian deep learning (variational inference of model parameters, section 3.2) is not motivated
- Overall, the approach is quite similar to the one presented in (Lienen and Günnemann, 2022) in terms of method-of-lines and interpolation of data, though the authors only mention but don't compare to them. In contrast to the author's writing in their related work section, I would even see this as the most closely related work.
- Application of method of lines is not novel as claimed in line 37-38, see (Iakovlev et al., 2020) and (Lienen and Günnemann, 2022)
- This paper does not actually propose/apply a collocation method as claimed in line 37-38
- As a result, novelty and significance of this work are limited

(Iakovlev et al., 2020): https://arxiv.org/abs/2006.08956
(Lienen and Günnemann, 2022): https://arxiv.org/abs/2203.08852

**Questions:**

- What are the advantages of modeling PDEs in a latent space instead of data space? Do you have any experimental insight into this question?

**Limitations:**

The authors did not discuss any limitations of their model.

---

> ### Author Rebuttal · Authors · 2023-08-08
>
> Thank you for your detailed review. We address the raised concerns below.
>
> **Motivating the use of generative modeling and Bayesian inference**
>
> We agree that we can improve our presentation regarding these points. Generative modeling is a standard approach in state space modeling which allows to explicitly define the parameter, process, and observation models, allowing to make explicit modeling assumptions, see e.g., "Statistics for Spatio-Temporal Data" by by N. Cressie. The ability to incorporate uncertainty in the predictions is valuable when there is limited amounts of data and in challenging real-world scenarios, where it allows to account for inherent system stochasticity and model uncertainty. Please note that we use the mean of the posterior predictive distribution as our prediction, which implies that we fully utilize the Bayesian nature of our method. We will add these clarifications to the revision.
>
> **Comparison to other methods and to Lienen and Günnemann, 2022**
>
> Indeed, the method by Lienen and Günnemann, 2022 is related to ours as it also models data on irregular spatiotemporal grids. However, as we show in the new set of experiments (see the global rebuttal), it fails in more realistic and challenging setting that we consider in our work.
>
> We agree that a more detailed comparison to a larger number of relevant methods is required. We added three more methods, a simple baseline NeuralODE [1], and two SOTA methods as suggested by the reviewers: Neural Stochastic PDEs (NSPDE) [2], and Finite Element Networks (FEN) from Lienen and Günnemann, 2022 [3]. Please see the global rebuttal for description of the setup and results
>
> **Collocation method and the method of lines**
>
> We agree that our method does not utilize the classical form of the collocation method. But please note that our method closely follows the collocation method, and then modifies it using the method of lines. In more detail, we represent z(t, x) as an interpolant (Eq. 3) and substitute it in the PDE (Eq. 4), which is then represented as a system of ODEs (Eq. 8), where each ODE corresponds to an observation location. This is very similar to how the collocation method is applied to time-dependent PDE problems. However, instead of using the collocation method directly, we combine it with the method of lines. While application of the method of lines is not novel, its combination with the collocation method, to the best of our knowledge, is novel and, as we discuss in lines 80-92, leads to multiple advantages such as grid-independence.
>
> We agree that the method of lines is not constrained to a particular type of spatial grids, but the way it was used in previous works makes their models grid-dependent. The main reason for grid-dependence, as discussed in lines 96-107, is reliance on the locations of the grid nodes, which our proposed methods avoids.
>
> **Questions**
>
> **Q1:** What are the advantages of modeling PDEs in a latent space instead of data space? Do you have any experimental insight into this question?
>
> **A1:** This is an important question as this is one of the main contributions of our work. We demonstrate the advantages of our latent PDE model in Fig. 7, where we compare a data-space variant of our model (latent state dimension d=1), and a latent space variant of our model (d > 1). As can be seen, modeling in the latent space results in drastic improvements in predictive performance. Setting d > 1 allows the encoder to infer the missing features of the state variable, which in turn allows the dynamics function to model the system more accurately. In other words, modeling a system in the data space sets a fundamental limitation to the model and its accuracy if the system states are only partially observed, such as in the Scalar Flow as well as in many real-world datasets.
>
> **References**
>
> [1] Neural Ordinary Differential Equations, https://arxiv.org/abs/1806.07366
>
> [2] Neural Stochastic PDEs: Resolution-Invariant Learning of Continuous Spatiotemporal Dynamics, https://arxiv.org/abs/2110.10249
>
> [3] Learning the Dynamics of Physical Systems from Sparse Observations with Finite Element Networks, https://openreview.net/forum?id=HFmAukZ-k-2

---

> > ### Comment · Reviewer_mL6m · 2023-08-13
> >
> > Thank you for the clarifications. I will remain with my assessment.

---

> > > ### Author Response · Authors · 2023-08-17
> > >
> > > Thank you for your comment. Please let us know if we can clarify any remaining concerns.

---

> > > > ### Comment · Reviewer_mL6m · 2023-08-19
> > > >
> > > > After further, careful consideration of all reviews combined, I increase my score to 5.

---

### Official Review · Reviewer_yPZT · 2023-07-05

**Soundness:** 3 good
**Presentation:** 2 fair
**Contribution:** 3 good
**Rating:** 6
**Confidence:** 4

**Summary:**

This paper introduces a new method for learning time-dependent PDE solutions with noisy, partially-observed data on irregular grids. This setting is quite challenging and is aligned with real-world data acquisition. They adopt a generative framework and combines two techniques for solving PDEs: the collocation method, which is used to approximate spatial derivatives, and the method of lines to propagate the PDE solution forward in time. At the core of their architecture lies a spatio-temporal encoder, that aggregates for every spatial/ temporal coordinate pair $(x_j, t)$ information over the spatial neighborhood $\mathcal{N}_S(x_j)$ and the temporal neighborhood $\mathcal{N}_T(t)$. They tackle the bayesian problem with a variational formulation, and learn the overall model by maximizing an ELBO objective. They use sparse bayesian multiple shooting to stabilize and accelerate the training. They validate their framework on three different datasets: Shallow-Water (SW), Navier-Stokes (NS) and Scalar Flow (SF).

**Strengths:**

This is overall a good technical contribution. The paper shows solid experimental results, outperforming DINo and MAgNet on the three different datasets. The idea of using a generative model for solving PDEs is interesting, and the combination of all the different blocks is far from trivial to train. The explanation of the architecture is clear, though the notations are sometimes a bit hard to follow. The authors also evaluate the capabilities of the model to adapt to coarser and finer grids to support their claims.


**Weaknesses:**

* The paper is quite difficult to read overall. The introduction and related work are very brief, especially for such a difficult topic. It would help the reader to add some background from the PDE Deep Learning literature in those sections. For instance, there is no mention of Neural Operators which has been a popular topic recently, and there is no clear explanation of the limitations of existing state-of-the-art methods. __N.B.__: DINo which tackles a very similar problem (except for noiseless data), is not cited in the introduction nor the related work but is still used as the main baseline. __N.B.2__: The Section 3.2 with the explanation of the multiple shooting framework is a bit confusing and does not add much in terms of explanation of the model. On the other hand, inference is only exposed in the Appendix and should be in the core of the paper.

* Though the idea of spatiotemporal encoding is sound, its realization does not appear to be very elegant and some limitations that could contradict the original claim of spatial continuity come to mind. What if the interpolation in the spatial neighborhood is not applicable ? Let's say you have an obstacle in the domain or you want to know the solution at the boundaries of your system, how can this work ? In all the figures the frames seem to contain only the convex hull of the set of points. This is worrying in terms of possible application to domains with complex geometries which are of main interest for people working in computational dynamics.

* The results shown in Figure 10 are very impressive, especially for Navier-Stokes (NS) and Shallow-Water (SW). However I wonder, how can the model generalize this well on new test initial conditions with only 2 training trajectories? In terms of MAE, the method is already at 0.03 and 0.07 with two training samples for SW and NS while it reaches 0.015 and 0.041 with 60 samples. Do you have a possible explanation for this phenomenon ?

* I understand that the main motivation of the paper is to develop a framework for partially-observed, noisy, irregularly-spaced observations, and therefore there are not a lot of suitable candidate baselines. Still, I think it would help readers understand the difficulty of the tasks to have a comparison with more classical neural operators and pde solvers on regular settings.


**Questions:**

* Did you try interpolant functions other than linear interpolation ? Similarly, have you tried the model with spherical data to see how the interpolation on a sphere behaves ?
* How does the model scale with the number of grid points ? ~1100 points for a 2D domain does not appear to be of really high-resolution. Is the method still applicable for instance with standard grids used in the community such as 64x64 or 64x128 points ?
* Is the method able to capture high-frequency patterns that occur in turbulent flows ?
* Is it possible that the proposed model overfitted the training horizon and as a result obtained better results than DINo and MAgNet ? This would be in line with the training-size analysis. How does the model extrapolate over the training horizon ?


**Limitations:**

There is no limitation section. The authors discuss briefly the difficulties for the method to scale with the numbers of points.

---

> ### Author Rebuttal · Authors · 2023-08-08
>
> Thank you for your detailed review. We address the raised concerns below.
>
> **Paper structure**
>
> Thank you for your suggestions regarding the structure of our manuscript. We will improve our work by incorporating them into the revision. We will move details about forecasting to the main text and will extend the introduction and related work to include neural operators and other relevant models. We will also elaborate on the limitations of existing SOTA methods.
>
> **Comparison to other methods**
>
> We agree that a more detailed comparison to a larger number of relevant methods is required. We added three more methods, a simple baseline NeuralODE [1], and two SOTA methods as suggested by the reviewers: Neural Stochastic PDEs (NSPDE) [2], and Finite Element Networks (FEN) [3]. Please see the global rebuttal for description of the setup and results.
>
> **Applications of our method on non-convex spatial domains**
>
> Our method can be applied on non-convex spatial domains without any modifications. Existence of obstacles is not a problem as long as the interpolation method is applicable (which is the case for many interpolation techniques such as k-NN, IDW, linear interpolation, piece-wise polynomial interpolation).
>
> The state of the system can be obtained at any point within the convex hull of the observation locations. One should expect that the number and positions of the observation locations are appropriate for the problem at hand. In that case it would be possible to obtain the state sufficiently close to the boundaries.
>
> **Data efficiency.**
>
> To achieve outstanding data-efficiency shown in Fig. 10 we utilize spatiotemporal locality of dynamical systems. Namely, we use the fact that derivatives in a PDE are defined locally (hence we need only local information to define the time rate of change), and assume that the latent state depends only on a local spatiotemporal neighborhood (as discussed in Section 4.2). These ideas are reflected in the design of our model which operates on local spatial and temporal neighborhoods instead of working on the whole grid, such as DINo for example.
>
> **Questions**
>
> **Q1:** Spherical grids and other interpolation methods.
>
> **A1:** Spherical grids are out of the scope of our work, but our method can potentially be applied to such data assuming that an appropriate interpolation method for spheres is used and that the spatial neighborhoods are appropriately defined on the sphere.
>
> We also investigated other interpolation techniques. We added two new interpolation methods: k-NN and inverse distance weighting, and tested them on spatial grids with different resolution. Please see the global rebuttal for description of the setup and results.
>
> **Q2:** Is the method able to capture high-frequency patterns that occur in turbulent flows ?
>
> **A2:** We did not consider that particular question, but as our experimental results suggest, our model is capable of learning complex flow-based phenomena.
>
> **Q3:** How does the model scale with the number of grid points ?
>
> **A3:** Since our model operates on each grid point, it scales linearly with the number of grid points. For reference, our model occupies around 6GB on GPU during training for shallow water dataset.
>
> **Q4:** How does the model extrapolate over the training horizon ?
>
> **A4:** To test this, we trained our model only on the first half of the training time grid (i.e., from 0 to 0.5T, where T is the observed time-horizon), and tested it on the full time grid (of length T). The test error for shallow water and scalar flow did not increase, while for Navier-Stokes it increased almost twice. This observation can be understood by the fact that velocity of the species transported in the Navier-Stokes dataset keeps increasing over time. Thus, the system behavior that the model was trained on (before time 0.5T), is different from the system behavior on which the model was tested (after time 0.5T), naturally leading to higher prediction errors.
>
>
> **References**
>
> [1] Neural Ordinary Differential Equations, https://arxiv.org/abs/1806.07366
>
> [2] Neural Stochastic PDEs: Resolution-Invariant Learning of Continuous Spatiotemporal Dynamics, https://arxiv.org/abs/2110.10249
>
> [3] Learning the Dynamics of Physical Systems from Sparse Observations with Finite Element Networks, https://openreview.net/forum?id=HFmAukZ-k-2

---

> > ### Comment · Reviewer_yPZT · 2023-08-14
> > **Answer to Rebuttal**
> >
> > > **Paper Structure**
> >
> > Thank you for clarifying the method's structure.
> >
> > > **Comparison to Other Methods**
> >
> > Appreciate the included baselines. While not precisely what I requested, they still enhance the experimental evaluation. I wanted to highlight the underlying complexity of the PDE without noise + irregular grid. Comparing against FNO or DeepOnet on regular grids could add value to your work.
> >
> > > **Applications on Non-Convex Spatial Domains**
> >
> > Regarding the use of an interpolant function to query the spatial neighborhood $\mathcal{N}_s$, could there be issues if neighbor points fall outside the domain due to obstacles?
> >
> > > **Data Efficiency**
> >
> > I understand the point about local modeling's inductive bias aiding close-to-true PDE learning. Then, is this bias from architecture components alone, or does the Bayesian framework also contribute? Could the same results be achieved without variational training?
> >
> > > **A1:**
> >
> > Thank you for incorporating additional interpolation methods.
> >
> > > **A2:**
> >
> > Noted.
> >
> > > **A3:**
> >
> > Could you specify the batch size that was used ?
> >
> > > **A4:**
> >
> > The inclusion of temporal extrapolation evaluation is appreciated.
> >
> > Overall, most of my concerns are addressed, and I'm inclined to raise my score to 6.

---

> > > ### Author Response · Authors · 2023-08-17
> > >
> > > Thank you for your positive comments. We address the raised concerns below.
> > >
> > > **Comparison to Other Methods**
> > >
> > > We forgot to mention that NSPDE is an operator-based method designed for irregular grids, so it is a more natural choice for our setup than e.g., FNO. Sorry for confusion!
> > >
> > > **Applications on Non-Convex Spatial Domains**
> > >
> > > This is a good point. We actually faced this problem with the Scalar Flow datasets. We found that marking the "out of domain" nodes by setting their value to -1 was sufficient (the observations were between 0 and 1). We missed this details in our manuscript, so we will add it to the revision.
> > >
> > > **Data Efficiency**
> > >
> > > The local inductive bias is only due to the design of the model components (encoder and dynamics function). One could use point estimation instead of Bayesian inference and still enjoy the data-efficiency aspect of our model.
> > >
> > > **Could you specify the batch size that was used?**
> > >
> > > The batch size was 1.

---

### Official Review · Reviewer_VsBY · 2023-07-05

**Soundness:** 3 good
**Presentation:** 3 good
**Contribution:** 3 good
**Rating:** 6
**Confidence:** 3

**Summary:**

The authors propose a grid independent model for learning PDEs from noisy experimental data. The proposed framework is probabilistic with an encoder that handles data efficiency and makes the solution grid independent. I think better differentiating from Ayed et al. and DINO would be helpful for the novelty contribution even though experimentally in these settings, the proposed model performs better.

**Strengths:**

- The problem setting is important to be able to learn PDEs from noisy and sparse observation data.
- The paper is well-written with a good introduction of PDEs in the intro section.
- Good overview of the related data-driven methods, such as Neural Operators (Li et al., 2021) and MeshGraphNets (Pfaff et al., 2021).
- Grid independent solvers are advantages of deep learning models over numerical methods. It is a nice idea to define a neighborhood rather than use the adjacent gridpoints.
- The proposed method is a nice application of the methods of lines.
- The problem statement is well-defined.
- It is good that the authors are considering the real-world setting of extrapolation in time.
- Adding probabilistic output is also important contribution.
- Synthetic and real-world datasets are both tested including a nice set of benchmark challenging datasets, e.g., Shallow water and Navier stokes. The scalar flow real-world camera dataset is also interesting.
- Nice experimental setting going from a given grid to a coarser one and vice vera to finer.
- The data efficiency results are nice.

**Weaknesses:**

- There is no guarantee that the PDEs learned from data will be better than the classical numerical PDE modeling so the first paragraph of the introduction could be modified a bit.
- The authors should better motivate the irregularly spaced noisy and partial observations in this context. In this case, Gaussian Processes or Attentive Neural Processes could also be considered as baselines. In particular "Learning Physical Models that Can Respect Conservation Laws" [Hansen et al., ICML 2023] uses a constrained generative ANP model to perform the interpolation task from noisy sparse data to a fine grid. This work also uses a Gaussian observation model as in Eqn. 9.
- There are advantages to operator-based methods vs. interpolation. I think a comparison to those methods, which are resolution-independent would be beneficial as well and should also be added to the related work section rather than just considering interpolation based methods.
- Small number of comparisons to only DINo and MAgNet as baselines. Neural Operator and MeshGraphNets could also be considered as baselines.
- The use of Neural ODEs is similar to the approach in "Continuous PDE Dynamics Forecasting with Implicit Neural Representations" [Yin et al., ICLR 2023]. In this work, the authors parameterize the latent space using implicit neural representations and then solve the ODE using Neural ODEs. It is good that the authors benchmark against this work in DiNO
- The GNN based models such as MeshGraphNet also assume the derivatives are computed using the neighboring points as mentioned on lines 97-99. While this is important to highlight, since the approach here uses a neighborhood rather than grid points, it may be better to move this to a related work section so the novelty and design of the proposed method is only discussed in 3.1.
- The model seems to be connecting various methods from different papers such as Neural ODEs and Iakovlev et al., 2023. It may be good to highlight and clarify the novelty in particular with respect to Iakovlev et al., 2023.
- It looks like several heuristics need to be applied to make Neural ODEs work here. Also see "Learning continuous models for continuous physics" [Krishnapriyan, 2022]. It may be simpler to use a discrete numerical time-stepping method as done in "CROM: Continuous Reduced-Order Modeling of PDEs Using Implicit Neural Representations" [Chen et al., ICLR 2023]. This is also used in the MAgNet baseline and I don't think the only changes in the time-stepping scheme from discrete Euler to Neural ODEs (also done in DiNO) is sufficient for the novelty for publication. Forward Euler is also unstable in many cases as shown in the aforementioned Krishnapriyan, 2022 and higher order schemes like Runge-Kutta 4 (RK4) may be beneficial.
- The overall model has many components and is quite complicated. I was wondering if any ablations were run to see which component is the most important.
- 3-4 hours depending on the problem size and architecture is a bit slow. I think cost-accuracy comparisons to numerical solvers in the experimental section would be beneficial.
- Figures 7-8 take significant space and may be moved to the appendix as ablations especially since they also show the results of the proposed method. The error plots between the predicted and exact may be easier to visualize.The scalar flow results look diffused.
- I think stating 'very high accurate predictions" on line 254 is a bit strong. The magnitude of the error is still high 1e-2 in comparison to numerical solvers
Minor
- For notation $u_i^j$ in numerical methods, the superscript typically notates time and subscript notates spatial coordinate.
- I would give Section 3 a more descriptive title than Method and also the name the proposed method as well.
- Can clarify on line 226 that the true latent space dimension for Navier-Stokes and Shallow water is 3 due to the x,y components of the velocity and the scalar pressure.
- Capital A on line 236
- "Similarly" to similar on line 240
- No parathesis in references on line 272.

**Questions:**

1. I know the authors mention that they use linear interpolation in the results in this paper and I was wondering if they performed any ablations with different types of interpolation.
2. It is a bit of strong assumption that the neighboring points are sufficient to compute all the derivatives. The authors mention finite differences here but higher order finite difference methods can use more grid points in the stencils than just the neighbors. Can higher-order stencils be considered here? It would also be good to add a reference for finite difference methods, e.g. LeVeque's numerical analysis book.
3. The authors mention that they test neighborhoods of various shapes and sizes. I think this is an important ablation. How ere two concentric circles chosen? The shapes are very important in numerical methods such as finite elements and those that use a Voronoi tessellation. In this case, it's a bit hard for me to see how it is purely grid independent. The number of evaluation points must also be carefully chosen.
4. Also in Equation 8, the ODEs are defined only at the gridpoints and then interpolation is performed at every time step in the ODE solver - how expensive is this?
5. Why is the distribution modeled as Gaussian and can the model be extended to support other distributions?
6. How is the temporal neighborhood size delta_T determined? Since the model performs extrapolation in time shouldn't this neighborhood only depend on the prior time steps?
7. Were simpler architectures than Transformers tested for the temporal encodings?
8. I understand that latent space dimension for the synthetic 2D datasets but for the scalar flow, how do we interpret the improvement in prediction quality for d=5 instead of d=1 despite similar metrics? Can you prove any relation between latent dimension d and the error since it seems experimentally to decay monotonically?

**Limitations:**

Limitations on the expense of the proposed method and future work are discussed.

---

> ### Author Rebuttal · Authors · 2023-08-07
>
> Thank you for your detailed review. We address the raised concerns below.
>
> **Comparison to numerical solvers**
>
> As discussed in Section 2, we assume the dynamics are unknown, hence numerical solvers are not applicable since they require fully known system dynamics.
>
> **Comparison to other methods**
>
> We added three more methods, a baseline NeuralODE [3], and two SOTA methods as suggested by the reviewers: Neural Stochastic PDEs (NSPDE) [4], and Finite Element Networks (FEN) [5]. Please see the global rebuttal for description of the setup and results.
>
> **Model complexity, importance of model components, and ablation studies**
>
> Our method consists of multiple components. Encoder is used to infer the latent state from observations. Dynamics function is used to propagate the latent state in time. Decoder maps the latent state to parameters of the observation distribution. And multiple shooting is a parameter inference technique which accelerates and stabilizes training. Considering the above, each component has its role and is important.
>
> Regarding various model parameters, we conducted multiple ablation studies throughout our work, please see the results in Fig. 7, 13, 14, and 15.
>
> **Discrete-time models and heuristics**
>
> In our setting the temporal grids are irregular, thus discrete-time models are not applicable. Training dynamic models is hard and often requires heuristics (one-step training, progressive increasing the length of the training trajectory, multiple shooting, and other) to stabilize training.
>
> **Novelty relative to other methods.**
>
> As outlined in the Introduction, the main advantage of our model are: (i) grid-independence, (ii) space-time continuity, (iii) data-efficiency, (iv) learning from partial observations, an (v) fast and stable training. No previous method has all these features.
>
> The main differences wrt Iakovlev et al., 2023 is that they do latent ODE modeling, while we do latent PDE modeling. Also, their model is not applicable in our case since: 1) The encoder and decoder are grid-dependent and applicable only on uniform spatial grids, 2) Their dynamics function could not be adapted to the PDE setting as it assumes that every point affects every other point on the grid, which is not the case in PDEs, where each point is affected only by points within a small neighborhood.
>
> **Paper structure**
>
> Thank you for your suggestions regarding the structure of the manuscript. We will improve our work by incorporating them into the revised version. Please see global rebuttal for details.
>
> **Questions**
>
> **Q1:** Different interpolation methods.
>
> **A1:** We investigated other interpolation techniques. We added two new interpolation methods: k-NN and inverse distance weighting, and tested them on spatial grids with different resolution. Please see the global rebuttal for description of the setup and results.
>
> **Q2:** Using neighborhood values to compute derivatives, and extension to higher-order stencils.
>
> **A2:** As shown in Fig. 3, we do not use only neighbor nodes' values, instead, we define fixed spatial neighborhoods whose values depend on other nodes as well (lines 97-107). In Appendix D we investigate the effect of using spatial neighborhoods of different shapes and sizes, so extension to higher-order stencils is straightforward.
>
> **Q3:** Spatial neighborhood shape and grid-independence.
>
> **A3:** Concentric circles shape was selected because it has better coverage than e.g., a cross or a square. As we show in Fig. 14, the choice of the neighborhood shape (number of circles) does not have a strong effect on the model's performance. Also note that the spatial neighborhoods have the same shape and do not depend on the observation locations, which makes our model grid-independent as we discuss in lines 97-107.
>
> **Q4:** How expensive is interpolation?
>
> **A4:** Interpolation amounts to matrix-vector multiplication where the matrix has dimensions Cn-by-n, where n is the number of nodes, and C is a small constant that depends on the spatial neighborhood shape. For our choices of the interpolation methods the matrix is extremely sparse, hence requires negligible memory to store and the matrix-vector product can be computed efficiently. Overall, it is one of the least expensive parts of the model.
>
> **Q5:** Why is the distribution modeled as Gaussian and can the model be extended to support other distributions?
>
> **A5:** The observation distribution is Gaussian as this is a reasonable choice in the absence of any system-specific requirements. Our model is agnostic to the choice of likelihood models, and our model can be easily extended to other observation distributions by asking the decoder to output parameters of that distribution (as in Eq. 9).
>
> **Q6:** Temporal neighborhood size and dependence only on prior time steps.
>
> **A6:** The temporal neighborhood size is a hyperparameter. Using only previous time points to infer the latent state would be similar to filtering, while using also the future time points is similar to smoothing. Smoothing tends to be more accurate (see e.g., Kalman filtering/smoothing).
>
> **Q7:** Were simpler architectures than Transformers tested for the temporal encodings?
>
> **A7:** We also considered continuous-time versions of RNNs, but they are much slower than Transformers.
>
> **Q8:** Improvement of the results for larger latent space dimension d.
>
> **A8:** For the scalar flow, improvement in prediction quality for larger d can be interpreted similarly to the synthetic datasets. Setting d=1 is not sufficient since the "true state" has higher dimension. Setting d > 1 allows the encoder to infer the missing states which allows the dynamics function to model the system more accurately. Proving any relationships, beyond simply extrapolating the points to larger d, is difficult due to complexity of the system.
>
> **References**
>
> [1] https://arxiv.org/abs/1806.07366
>
> [2] https://arxiv.org/abs/2110.10249
>
> [3] https://openreview.net/forum?id=HFmAukZ-k-2

---

> > ### Comment · Reviewer_VsBY · 2023-08-14
> > **Response to Rebuttal**
> >
> > Thank you very much for your detailed rebuttal and additional experiments run with more baselines. The ablation on the different interpolation methods is also very interesting. I have the remaining three points:
> > - I do agree with Reviewer 1peH that theory and convergence analysis is very important and parts of the method are very heuristic. I don't think this statement "theory to guide all our choices, but unfortunately it is not always available for neural network parameterized models, and what remains is heuristics and empirical evidence." is a strong rebuttal by the authors since we need strong theory to justify and motivate these SciML works.
> > - I don't understand why discrete time models are not possible with irregular time steps. There are discrete adaptive time-stepping methods.
> > - It is good that the authors added additional baselines but they did not address my question on operator based method, such as FNO, which was also raised by Reviewer yPZT on the advantages of the proposed type of interpolation method vs. operator methods that are also resolution independent.
> >
> > For now, I will be keeping my score, thank you.

---

> > > ### Comment · Reviewer_VsBY · 2023-08-15
> > > **Rebuttal Followup**
> > >
> > > Based on the additional ablations on the interpolation, I will raise my score to 6, thank you.

---

> > > > ### Author Response · Authors · 2023-08-17
> > > >
> > > > Thank you for your positive comments. We address the raised concerns below.
> > > >
> > > > **We need strong theory to justify and motivate SciML works**
> > > >
> > > > We acknowledge the importance of theoretical convergence guarantees for some applications. At the same time we would like to highlight that the main contributions of our work are related to probabilistic and deep generative modeling techniques that we have developed for latent-space neural PDEs.
> > > >
> > > > **Comparison to an operator based method.**
> > > >
> > > > We forgot to mention that NSPDE model is operator-based. Sorry for the confusion! Furthermore, to compare super-resolution performance of our model to the new baselines we will add extra super-resolution experiments (as in Fig.8) with these baselines to the next revision.
> > > >
> > > > **Why discrete time models are not possible with irregular time steps?**
> > > >
> > > > We agree that discrete time models are also possible, but let us clarify our initial answer. Since discrete-time models are defined in terms of a map F from current to the next state, one could add the step size dt as the input to F (as was done e.g. in CROM). This is a feasible solution, but the model has to learn how to advance the state forward in time by dt and it works only for dts similar to those in the training data. Using a continuous-time model is a more natural alternative. We would also like to highlight that both discrete- and continuous-time models suffer from training instabilities and require stabilization methods (such as multiple shooting) [1, 2].
> > > >
> > > > **References**
> > > >
> > > > [1] On the smoothness of nonlinear system identification, https://arxiv.org/pdf/1905.00820.pdf
> > > >
> > > > [2] Gradients are Not All You Need, https://arxiv.org/pdf/2111.05803.pdf

---

### Official Review · Reviewer_1peH · 2023-07-06

**Soundness:** 2 fair
**Presentation:** 3 good
**Contribution:** 2 fair
**Rating:** 3
**Confidence:** 5

**Summary:**

Modeling systems with spatiotemporal evolutions, such as the ones arise in problems governed by PDEs, is challenging. This is more pronounced when the system's underlying mechanisms are too complex or unknown. The authors propose a grid-independent generative model from noisy and partial observations on irregular domains. The latent state dynamics is constructed by merging the ideas inspired by the classical numerical PDE analysis such as collocation method and the method of lines, which is discussed in section 3.1 and 3.2. They also propose a novel encoder design that operates on local spatiotemporal neighborhoods for improved data-efficiency and grid-independence. The authors apply that their model three use cases from two synthetic analysis and one from real-world datasets.

**Strengths:**

Multiple shooting analysis for training datasets from dynamical systems with long time simulations is a creative idea to enable reasonable cost of learning and avoid learning instabilities.
In Appendix D, they analyzed the impact of radius and multiple shooting on the overall performance of the model to further complete the analysis.

The examples provided are sophisticated enough to show that the framework can be applied to even more realistic scenarios.


**Weaknesses:**

I have two major concerns about this work:
1.	There are various heuristic choices/assumptions with no concrete proofs. For example, the choice of linear interpolant to map the values at the grid points of z(t) to z(t,x) is arbitrary. Moreover, when authors write the dynamical system in form of equation 7, while they are inspired by collocation methods, they are not actually utilizing such method. Instead, they argue that the functions and its (sufficiently smooth) derivatives are a function of neighborhood locations. This is another heuristic assumption which prevents a careful convergence analysis. Can authors prove if the radius tends to zero, such solution converges to the “real” solution? In what order the error decreases? The analysis in Appendix D is in fact anecdotal and does not mathematically guarantee a desired performance. Finally, regarding multiple shooting method, while I think this is a very nice idea, it’s not novel. For example, see "50 years of time parallel time integration." By Martin, where several other methods other than multiple shooting are proposed to accelerate ODE integration. While I don’t see any fundamental error, I am not convinced that the analysis is mathematically sounds for computational adoption.
2.	The topic of the paper is closely related to model discovery, which has been investigated in PDE community for quite some time now. A more appropriate comparison would be the one with SINDy ("Discovering governing equations from data by sparse identification of nonlinear dynamical systems." By Brunton et al), Neural ODE (Neural Ordinary Differential Equations by Chen et al), or auto-encoder-based discussions (Data-driven discovery of coordinates and governing equations by Champion et al). Please also note, recently people also used neural ODE with collocation points (Physics-informed neural ODE (PINODE): embedding physics into models using collocation points by Sholokhov et al). To better position this work in the larger community, a much more detailed comparison is needed.


**Questions:**

The fact that the number and positions of sensor (observation locations) change with time may actually help the problem at the time of inference (Reconstruction). Have authors considered a fixed setting for which the observation locations are the same? Is there a minimum number of sensors required for a certain accuracy? Are there any insights about using this approach for sensor placement problems?

In equation 3, authors propose using a linear interpolation: what about other choices of interpolant? Any studies show performance of other choices? Seems to me that a linear interpolant may require a larger number of observation locations? Any insights?

In equation 7, what is the order of such analysis? Can authors prove this converges to actual dynamics? Seems heuristic. By order, I mean the order of accuracy of the approximation of the derivative in terms of the distance to neighborhood points (big O s^n).

In equations 12 and 13, what is the implication of assuming that the continuity inducing prior is Gaussian? Such assumption makes the solution of the problem computationally tractable but what if the actual process is non-Gaussian? What are limitations?

In the results section, for shallow water the model for d=5 (and maybe even 4) outperforms fully observed MAE which is based on true d=3. Are there any insight for this?

Minor comment: the notation for $\partial z/\partialx_x$ on line 56 and elsewhere with a dot is a bit obscure. Maybe more common choice for notation is to use n as the degree of differentiation.
Also, A is capitalized on line 236.

---

> ### Author Rebuttal · Authors · 2023-08-07
>
> Thank you for your detailed review. We address the raised concerns below.
>
> **Heuristics**
>
> We would be glad to rely on theory to guide all our choices, but unfortunately it is not always available for neural network parameterized models, and what remains is heuristics and empirical evidence.
>
> An interpolation method is often chosen based on a modeling decision or empirical evidence on its performance. We chose linear interpolation since for our case it provides sufficient accuracy and computationally efficiency. However, your comment inspired us to investigate other interpolation techniques. We added two new interpolation methods: k-NN and inverse distance weighting, and tested them on spatial grids with different resolution. Please see the global rebuttal for description of the setup and results.
>
> As you correctly noted, we do not use the classical form of the collocation method. Instead, we modify it by combining it with the method of lines (see Section 3.1 for motivation). Based on intuition from finite difference formulas, using a universal neural network based function approximator and decreasing the spatial neighborhood size should improve the accuracy of derivative estimation and hence the predictive performance. However, in practice the best performance is achieved with relatively large neighborhood sizes (see Appendix, Fig. 13). The optimal neighborhood size depends on the system being modeled. Overall, this is an important part of our proposed model that guarantees good performance and efficient computation.
>
> Please note that we do not use classical multiple shooting and do not optimize over the shooting parameters. Instead, we introduce an encoder that maps observations directly to shooting states, and optimize over parameters of the encoder. As the result, the number of optimization parameters does not grow with the dataset size and spatiotemporal resolution, and no separate optimization loop is required at test time to infer the shooting states. Furthermore, Fig. 15 in Appendix shows that adoption of multiple shooting is worthwhile as it reduces training time and improves predictive performance.
>
> **Comparison to other methods**
>
> We agree that a more detailed comparison to a larger number of relevant methods is required. We added three more methods, a simple baseline NeuralODE [3], and two SOTA methods as suggested by the reviewers: Neural Stochastic PDEs (NSPDE) [4], and Finite Element Networks (FEN) [5]. Please see the global rebuttal for description of the setup and results.
>
> **Questions**
>
> **Q1:** Number and positions of sensors.
>
> **A1:** Please note that we always use fixed observation locations (we will clarify this point in the revision). Studying optimal sensor placement is outside the scope of our work. However, moving observation locations could indeed be beneficial for the model's performance, especially if they are moved to parts of the space where more accurate resolution is required.
>
> **Q2:** Equation 3, other interpolation methods and number of observation locations.
>
> **A2:** We added results for other interpolation methods (see above). Other study ([1], Fig. 6) compared different types of interpolants, including learned ones, and showed little difference relative to the linear interpolation. Given a fixed number of observation locations, different interpolation techniques have different trade-offs between accuracy and computational efficiency. Linear interpolation is more on the efficiency side, but as we show in Fig. 8 and Table 1 above it performs well even on very coarse spatial grids.
>
> **Q3:** Equation 7 and convergence to true dynamics.
>
> **A3:** The fact that our model makes accurate predictions implies that the dynamics it learns are close to the true dynamics of the system. Giving theoretical convergence guarantees is challenging due to complexity of the model.
>
> **Q4:** Implication of assuming that the continuity inducing prior is Gaussian.
>
> **A4:** The purpose of the continuity prior is to bring the shooting and the system states close to each other. Assuming Gaussian prior implies that closeness is measured in terms of the squared distance. Note that the underlying process that all neural PDE methods aim to learn is a continuous-time deterministic PDE, and the sole motivation of multiple shooting is to introduce an approximation that converts the original problem into a form that enables efficient and robust optimization. In other words, the choice of the continuity prior corresponds to a choice of a multiple shooting approximation.
>
> **Q5:** Model performance for different latent state dimensions.
>
> **A5:** Based on our experience, the best performance in latent-state dynamic models is achieved for latent space dimensions larger than the true system state dimension. Similar observations were made in other works, e.g., [2]. A possible explanation for this is that in larger latent spaces simpler dynamics might be sufficient to model the system's behavior.
>
>
> **References**
>
> [1] MAgNet: Mesh Agnostic Neural PDE Solver, https://arxiv.org/abs/2210.05495
>
> [2] Non-linear State-space Model Identification from Video Data using Deep Encoders, https://arxiv.org/abs/2012.07721
>
> [3] Neural Ordinary Differential Equations, https://arxiv.org/abs/1806.07366
>
> [4] Neural Stochastic PDEs: Resolution-Invariant Learning of Continuous Spatiotemporal Dynamics, https://arxiv.org/abs/2110.10249
>
> [5] Learning the Dynamics of Physical Systems from Sparse Observations with Finite Element Networks, https://openreview.net/forum?id=HFmAukZ-k-2

---

### Author Rebuttal · Authors · 2023-08-08

We would like to thank all reviewers for their careful reading and detailed comments. We believe that the suggested revisions have improved our manuscript. We believe our answers address all review comments, but if anything remains unclear we are happy to provide further clarifications.

We provided a response for each reviewer individually, and use this section to provide details and results of the extra experiments that the reviewers requested.

**Comparison to other models**

We agree that a more detailed comparison to a larger number of relevant methods is required. We added three more methods, a simple baseline NeuralODE [1], and two SOTA methods as suggested by the reviewers: Neural Stochastic PDEs (NSPDE) [2], and Finite Element Networks (FEN) [3]. We describe the setup and results below.

Setup: In addition to our partially-observed datasets, we created fully-observed versions of the synthetic datasets, where the whole system state is observed. This is a simplified setup where most models show good results. In the table below we show test MAE of all the methods.

|  Model | Shallow Water (Full) | Shallow Water (Partial) | Navier Stokes (Full) | Navier Stokes (Partial) | Scalar Flow (Real-World) |
|:------|:--------------------:|:-----------------------:|:--------------------:|:-----------------------:|:------------------------:|
| NODE   |   $0.036 \pm 0.000$  |    $0.084 \pm 0.001$    |   $0.053 \pm 0.001$  |    $0.109 \pm 0.001$    |     $0.056 \pm 0.001$    |
| FEN    |   $\boldsymbol{0.011 \pm 0.002}$  |    $0.064 \pm 0.005$    |   $0.031 \pm 0.001$  |    $0.108 \pm 0.002$    |     $0.062 \pm 0.005$    |
| NSPDE  |   $0.019 \pm 0.002$  |    $0.033 \pm 0.001$    |   $0.042 \pm 0.004$  |    $0.075 \pm 0.002$    |     $0.059 \pm 0.002$    |
| DINo   |   $0.027 \pm 0.001$  |    $0.063 \pm 0.003$    |   $0.047 \pm 0.001$  |    $0.113 \pm 0.002$    |     $0.059 \pm 0.001$    |
| MAgNet |          NA          |    $0.061 \pm 0.001$    |          NA          |    $0.103 \pm 0.003$    |     $0.056 \pm 0.003$    |
| Ours   |   $0.014 \pm 0.002$  |    $\boldsymbol{0.016 \pm 0.002}$    |   $\boldsymbol{0.024 \pm 0.003}$  |    $\boldsymbol{0.041 \pm 0.003}$    |     $\boldsymbol{0.042 \pm 0.001}$    |

We see that some of the baseline models achieve reasonably good results on the fully-observed datasets, but they fail on partially-observed data, while our model maintains strong performance in all cases. Apart from the fully observed shallow water dataset where FEN performs slightly better than ours, our proposed method outperforms other methods on all other datasets by a clear margin. We will add these results into the revised manuscript.

**Investigating other interpolation methods**

We investigated other interpolation techniques and describe the setup and results below.

Setup: We used spatial grids with three different resolutions (Coarser, Original, and Finer). We trained our model only on the Original grid, and tested it on all grids to test how well it generalizes to grids with lower/higher resolution. We added two new interpolation methods: k-NN and inverse distance weighting. The results (test MAE) are shown in the table below.

|    Dataset    |   Grid   |        k-NN       |       Linear      |         IDW        |
|:-------------:|:--------:|:-----------------:|:-----------------:|:------------------:|
|               | Coarser  | $0.046 \pm 0.002$ | $0.034 \pm 0.001$ | $ 0.038 \pm 0.002$ |
| Shallow Water | Original | $0.017 \pm 0.002$ | $0.016 \pm 0.002$ |  $0.017 \pm 0.003$ |
|               | Finer    | $0.031 \pm 0.003$ | $0.017 \pm 0.003$ |  $0.030 \pm 0.002$ |
|               | Coarser  | $0.087 \pm 0.006$ | $0.069 \pm 0.009$ |  $0.066 \pm 0.006$ |
| Navier Stokes | Original | $0.048 \pm 0.009$ | $0.041 \pm 0.003$ |  $0.045 \pm 0.010$ |
|               | Finer    | $0.054 \pm 0.009$ | $0.044 \pm 0.004$ |  $0.049 \pm 0.002$ |
|               | Coarser  | $0.041 \pm 0.021$ | $0.032 \pm 0.009$ |  $0.035 \pm 0.012$ |
| Scalar Flow   | Original | $0.019 \pm 0.001$ | $0.018 \pm 0.000$ |  $0.018 \pm 0.001$ |
|               | Finer    | $0.040 \pm 0.016$ | $0.026 \pm 0.006$ |  $0.028 \pm 0.007$ |

We see that all interpolation methods perform rather similarly on the Original grid, but linear interpolation and IDW tend to perform better on finer/coarser grids than k-NN. Since our method can be combined with essentially any interpolation technique, this leaves a user the freedom to choose a particular technique that works for his/her application, although based on our experiments linear interpolation is often accurate. We believe this ablation is a valuable addition to our manuscript and we will include these experiments together with general discussion on choosing an interpolation method into the revised manuscript.

**References**

[1] Neural Ordinary Differential Equations, https://arxiv.org/abs/1806.07366

[2] Neural Stochastic PDEs: Resolution-Invariant Learning of Continuous Spatiotemporal Dynamics, https://arxiv.org/abs/2110.10249

[3] Learning the Dynamics of Physical Systems from Sparse Observations with Finite Element Networks, https://openreview.net/forum?id=HFmAukZ-k-2

---

### Decision · Program_Chairs · 2023-09-21

**Decision:**

Accept (poster)

**Comment:**

The paper presents a method for solving Partial Differential Equations (PDEs) using partially observed data on irregular spatio-temporal grids. This approach combines techniques inspired by collocation and the method of lines. Notably, it introduces a spatio-temporal encoder that aggregates neighbor information, coupled with a variational formulation. The method's performance is assessed through comparisons with various baselines on three datasets of trajectories, each corresponding to a distinct PDE generated from simulations.

All the reviewers agree on the importance of the problem addressed in the paper and acknowledge the strong performance of the proposed model. They conducted thorough analyses of the paper, including questions about its originality (as it seems to build upon [Iakovlev et al., ICLR 2023]), the initial experimental evaluation that relied on only two baselines, and the various heuristics employed in the model.

In response to these comments, the authors provided comprehensive answers and conducted additional experiments, including the inclusion of three new baselines. These new experiments reinforced the initial findings of the paper. The reviewers found these additions to be valuable and were convinced by the responses.

The authors are strongly encouraged to consider the reviewers' comments in order to enhance the organization and clarity of the paper